# Unlearn and Burn: Adversarial Machine Unlearning Requests Destroy Model Accuracy

**Yangsibo Huang**[1,2*]  **Daogao Liu**[3*]  **Lynn Chua**[1]  **Badih Ghazi**[1]  **Pritish Kamath**[1]
**Ravi Kumar**[1]  **Pasin Manurangsi**[1]  **Milad Nasr**[1]  **Amer Sinha**[1]  **Chiyuan Zhang**[1]
[1]Google  [2]Princeton University  [3]University of Washington

## ABSTRACT

Machine unlearning algorithms, designed for selective removal of training data from models, have emerged as a promising approach to growing privacy concerns. In this work, we expose a critical yet underexplored vulnerability in the deployment of unlearning systems: the assumption that the data requested for removal is always part of the original training set. We present a threat model where an attacker can degrade model accuracy by submitting adversarial unlearning requests for data *not* present in the training set. We propose white-box and black-box attack algorithms and evaluate them through a case study on image classification tasks using the CIFAR-10 and ImageNet datasets, targeting a family of widely used unlearning methods. Our results show extremely poor test accuracy following the attack—3.6% on CIFAR-10 and 0.4% on ImageNet for white-box attacks, and 8.5% on CIFAR-10 and 1.3% on ImageNet for black-box attacks. Additionally, we evaluate various verification mechanisms to detect the legitimacy of unlearning requests and reveal the challenges in verification, as most of the mechanisms fail to detect stealthy attacks without severely impairing their ability to process valid requests. These findings underscore the urgent need for research on more robust request verification methods and unlearning protocols, should the deployment of machine unlearning systems become more prevalent in the future.

## 1 INTRODUCTION

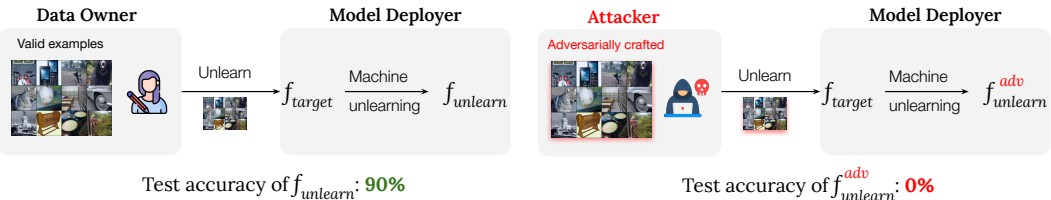

Figure 1: Machine unlearning allows data owners to remove their training data from a target model without compromising the unlearned model's accuracy on examples not subject to unlearning requests, such as test data (left). However, we demonstrate that adversarially crafted unlearning requests, though visually similar to legitimate ones, can lead to a catastrophic drop in model accuracy after unlearning (right).

Machine unlearning (e.g., Cao & Yang, 2015; Garg et al., 2020; Cohen et al., 2023) is a notion formulated to address a critical challenge in contemporary machine learning systems: the selective removal from trained models of information pertaining to a given subset of training examples. This capability has become increasingly important as machine learning models are frequently trained on large datasets, which often unintentionally include private (Carlini et al., 2021) and copyrighted (Henderson et al., 2023; Lee et al., 2024; He et al., 2024; Wei et al., 2024) material. The need to "unlearn" specific data points is not merely a technical challenge; it is also a response to escalating privacy concerns and evolving legal frameworks, such as the General Data Protection Regulation (GDPR, European Parliament & Council of the European Union).

---

*Equal contribution (yangsibo@google.com, dgliu@uw.edu). The authors after the first two are listed in alphabetical order. Code is available at https://github.com/daogaoliu/unlearning-under-adversary.

At its core, the goal of machine unlearning is to provide a protocol for data owners to request the removal of their data from a model. Specifically, let $\mathcal{D}_{\text{train}}$ be the training set, and $f_{\text{target}} := \mathsf{L}(\mathcal{D}_{\text{train}})$ be a model returned by the learning algorithm $\mathsf{L}$. A machine unlearning algorithm $\mathsf{U}$ takes the trained model $f_{\text{target}}$ and a *forget set* $\mathcal{D}_{\text{forget}} \subset \mathcal{D}_{\text{train}}$, and produces a new model $f_{\text{unlearn}} := \mathsf{U}(f_{\text{target}}, \mathcal{D}_{\text{forget}})$ that is not influenced by $\mathcal{D}_{\text{forget}}$. The most straightforward unlearning algorithm is to retrain from scratch, i.e., to compute $f_{\text{retrain}} := \mathsf{L}(\mathcal{D}_{\text{train}} \setminus \mathcal{D}_{\text{forget}})$. However, this is generally impractical not only because it requires saving an entire copy of $\mathcal{D}_{\text{train}}$, but also because the cost of training from scratch has become prohibitively expensive for many modern neural network models. As a result, there have been numerous efforts in the literature proposing novel unlearning algorithms (Ginart et al., 2019; Liu et al., 2020; Wu et al., 2020; Bourtoule et al., 2021; Izzo et al., 2021; Gupta et al., 2021; Sekhari et al., 2021; Ye et al., 2022), some of which achieve better practical efficiency by relaxing the strong guarantees required in exact unlearning. This has led to extensive research and a continuous evolution of the formulation of the unlearning protocol that can better accommodate such considerations.

**Our contributions.** In this paper, we continue this line of research by systematically studying one critical component of the current unlearning protocol. Specifically, the interface of an unlearning algorithm $\mathsf{U}$ is defined as a mapping from $(f_{\text{target}}, \mathcal{D}_{\text{forget}})$ to $f_{\text{unlearn}}$, where it is assumed that $\mathcal{D}_{\text{forget}} \subset \mathcal{D}_{\text{train}}$. However, it is rarely considered (Q1) *what could go wrong when the assumption is violated (i.e., $\boldsymbol{\mathcal{D}_{forget}} \not\subset \boldsymbol{\mathcal{D}_{train}}$), and (Q2) is it possible to robustly verify this assumption via defensive mechanisms.* Both of these questions have significant implications for the real-world deployment of unlearning systems; addressing them will reveal potential risks of current designs. We evaluate various ways to improve the unlearning protocol that can potentially mitigate these risks. Specifically, we formulate a threat model and propose concrete attack algorithms in § 2; we study Q1 in § 3; and study Q2 in § 4. To summarize, our main contributions are:

- We identify a critical assumption ($\mathcal{D}_{\text{forget}} \subset \mathcal{D}_{\text{train}}$) in machine unlearning protocols that is generally overlooked in the literature, and formulate a threat model (§ 2.1) where an attacker can compromise model performance by exploiting this assumption, i.e., submitting unlearning requests that do not belong to the original training set.
- Inspired by meta-learning (Finn et al., 2017), we propose an attack algorithm (§ 2.2) to identify adversarial perturbations of the forget set examples by computing gradients through the unlearning updates using Hessian-vector Product (Dagréou et al., 2024). We further extend our algorithm to the black-box setting with zero-th order gradient estimation.
- To answer Q1, we evaluate our attack algorithms on three mainstream unlearning algorithms for image classification tasks with the state-of-the-art base models on the CIFAR-10 and ImageNet datasets (§ 3). Our results show extremely poor test accuracy following the attack—3.6% on CIFAR-10 and 0.4% on ImageNet for white-box attacks, and 8.5% on CIFAR-10 and 1.3% on ImageNet for black-box attacks. Moreover, we show that the identified adversarial forget sets can sometimes transfer across models.
- To answer Q2, we consider attack algorithms with an additional stealth objective that submit unlearning requests that are only slightly perturbed from true training images. We evaluate six verification schemes and found that none of them can effectively filter out malicious unlearning requests without severely compromising the ability to handle benign requests (§ 4).
- We provide discussion and ablation studies of different variations of threat models (§ 5).
- We present theoretical insights for the proposed attack (§ C), where we construct a formal setting of unlearning for linear models, and prove the existence of an attack wherein the unlearned model on a $\mathcal{D}_{\text{forget}}$ set consisting of slightly perturbed examples misclassifies all examples in $\mathcal{D}_{\text{retain}}$.

Overall, our findings highlight the pressing need for stronger request verification schemes in machine unlearning protocols, especially as their real-world deployment may become more prevalent in the future.

## 2    MACHINE UNLEARNING WITH ADVERSARIAL REQUESTS

In this section, we formulate a threat model (§ 2.1) in which the violation of the assumption $\mathcal{D}_{\text{forget}} \subset \mathcal{D}_{\text{train}}$ is not verified when fulfilling an unlearning request, and propose attack methods that generate malicious unlearning requests (§ 2.2).

## 2.1 THREAT MODEL

We focus on a data owner-side adversary who submits unlearning requests with the intent of causing performance degradation in the unlearned model. The adversary's goal and capabilities are detailed below.

**Adversary's goal.** The attacker aims to generate a set $\mathcal{D}^{\text{adv}}_{\text{forget}}$ of strategically crafted examples with a crucial property that we refer to as *performance-degrading*: After undergoing the unlearning process, these examples result in an unlearned model, $f^{\text{adv}}_{\text{unlearn}}$, with significantly worse performance (measured by metrics like accuracy in classification tasks) compared to a non-maliciously generated unlearned model $f_{\text{unlearn}}$, when evaluated on data not intended for removal, such as the retain set $\mathcal{D}_{\text{retain}}$ or any held-out dataset $\mathcal{D}_{\text{holdout}}$. The performance degradation can occur in two ways:

- *General*, where the degradation affects all examples.
- *Targeted*, similar to a backdoor attack (Chen et al., 2017), where the model's performance is degraded on a specific subset of examples (e.g., a specific class in classification tasks).

Optionally, another potentially desirable property is being *stealthy*, meaning that the crafted examples closely resemble valid training data. This would make it harder for model deployers to detect without implementing careful detection mechanisms. We leave consideration of the stealthy property to the discussion of detection mechanisms in § 4.

**Threat model.** We assume the attacker has access to a subset of the training data, $\mathcal{D}_{\text{train}}$[1]. We also assume that the attacker knows the unlearning algorithm used by the model. In § 5, we discuss relaxing both assumptions and demonstrate that the attack can still be very effective even if the attacker lacks access to $\mathcal{D}_{\text{train}}$ or lacks full knowledge of the unlearning algorithm.

Regarding the attacker's knowledge of or access to the model, we consider two settings:

- *White-box:* The attacker has full access to the model, allowing them to perform back-propagation through the model.
- *Black-box:* The attacker has query-only access to the model's loss on a set of arbitrarily chosen examples, without knowledge of the model weights or architecture.

## 2.2 WHITE-BOX AND BLACK-BOX ATTACK METHODS

The attacker's goal is to identify an adversarial forget set $X^{\text{adv}}_{\text{forget}}$ that, when fed into the unlearning algorithm, could maximize the degradation of the unlearned model's performance on the retain set. We propose an attacking framework to find $X^{\text{adv}}_{\text{forget}}$ with gradient-based local search. Specifically, let $g(X^{\text{adv}}_{\text{forget}}) := \mathcal{L}_{\text{retain}}(\mathsf{U}(f_{\text{target}}, \{X^{\text{adv}}_{\text{forget}}, y_{\text{forget}}\}, \mathcal{D}_{\text{retain}}))$ denote the retain loss after we run the unlearning algorithm $\mathsf{U}$ with $X^{\text{adv}}_{\text{forget}}$ as the forget set. We run gradient ascent on $g(X^{\text{adv}}_{\text{forget}})$ to identify the (local) optimal $X^{\text{adv}}_{\text{forget}}$. The main challenge is to compute gradient through the execution of the unlearning algorithm $\mathsf{U}$. In the following, we describe how to do this in two different access models.

**White-box attack.** With full access to the model, a white-box attacker can (hypothetically) construct a computation graph that realizes the unlearning update $\mathsf{U}$ to the underlying model, and compute the gradient $\nabla g(X^{\text{adv}}_{\text{forget}})$ by back-propagating through this computation graph. Note that many common unlearning algorithms $\mathsf{U}$ are implemented with gradient updates as well, therefore this hypothetical computation graph unrolling can be realized via *gradient-through-gradient*, which is supported by most modern deep learning toolkits[2] with the auto differentiation primitive of *Hessian-vector Products* (Dagréou et al., 2024). We note that a similar technique has been used in meta-learning algorithms such as MAML (Finn et al., 2017). However, the setup is quite different as MAML operates in the *weight* space, while we search in the model *input* space. Specifically, MAML tries to identify the best base model that could minimize the target loss when a fine-tuning algorithm is executed, whereas our algorithm tries to identify the best unlearning inputs that could maximize the target loss when an unlearning algorithm is executed, based on the unlearning inputs.

---

[1]This is a reasonable assumption, as it would naturally apply if the attacker is a data owner who has already contributed to training.

[2]In this paper, we use the higher library designed for PyTorch to implement this computation.

A formal description is shown in Algorithm 1. For simplicity, we initialize the gradient-based attack with a valid forget set (Line 2 in Algorithm 1). However, as we later demonstrate in § 5, the attacker could start with an arbitrary collection of datapoints, even random pixels.

---

**Algorithm 1** White-box attack.

---

1: **Input:** original model $f_{\text{target}}$, a collection $X \subset \mathcal{D}_{\text{train}}$ of training examples, retain set $\mathcal{D}_{\text{retain}}$, unlearning method $\mathsf{U}$, attack step size $\eta_{\text{adv}}$, optimizing steps $T_{\text{adv}}$, access to loss function $g(X_{\text{forget}}^{\text{adv}}) :=$ $\mathcal{L}_{\text{retain}}(\mathsf{U}(f_{\text{target}}, \{X_{\text{forget}}^{\text{adv}}, y_{\text{forget}}\}, \mathcal{D}_{\text{retain}}))$, i.e., the loss over $\mathcal{D}_{\text{retain}}$ after unlearning with $X_{\text{forget}}^{\text{adv}}$.
2: Initialize $X_{\text{forget}}^{\text{adv}} \leftarrow X$
3: **for** $t = 1 \rightarrow T_{\text{adv}}$ **do**
4:     Compute gradient $\nabla g(X_{\text{forget}}^{\text{adv}})$                   /* Using the gradient-through-gradient technique */
5:     $X_{\text{forget}}^{\text{adv}} \leftarrow X_{\text{forget}}^{\text{adv}} + \eta_{\text{adv}} \nabla g(X_{\text{forget}}^{\text{adv}})$        /* Update adversarial forget set to maximize retain loss */
6: **end for**
7: **Return:** $(X_{\text{forget}}^{\text{adv}}, y_{\text{forget}})$

---

**Black-box attack.** In the black-box setting, the attacker can only access the value of the loss $g(X_{\text{forget}}^{\text{adv}})$ and cannot directly compute its gradients. Consequently, instead of computing the gradient explicitly, we use a gradient estimator from the zeroth-order optimization literature (Duchi et al., 2015; Nesterov & Spokoiny, 2017), namely, we draw random noise $\Delta$ (of unit length for simplicity), and compute an estimate of the gradient as:

$$\nabla g(X_{\text{forget}}^{\text{adv}}) \approx \frac{g(X_{\text{forget}}^{\text{adv}} + \Delta) - g(X_{\text{forget}}^{\text{adv}} - \Delta)}{2}\Delta.$$

Furthermore, we implement the following heuristic to improve the local search: (i) Instead of directly optimizing for $X_{\text{forget}}^{\text{adv}}$, given a benign forget set $X$, we search for an adversarial perturbation $\mathbf{z}$ and let $X_{\text{forget}}^{\text{adv}} = X + \mathbf{z}$; (ii) We simultaneously optimize $m$ different randomly initialized adversarial perturbations and return the best one at the end of the algorithm; (iii) In each step, we independently sample $p$ random $\Delta$'s for gradient estimation for each of the $m$ perturbations. This results in $mp$ updated perturbations (some bad ones are removed, Line 18 of Algorithm 2), and we remove all but the top-$m$ of them at the end of the step. The full description is shown in Algorithm 2.

---

**Algorithm 2** Black-box attack.

---

1: **Input:** original model $f_{\text{target}}$, a collection $X \subset \mathcal{D}_{\text{train}}$ of training examples, retain set $\mathcal{D}_{\text{retain}}$, unlearning method $\mathsf{U}$, attack step size $\eta_{\text{adv}}$, optimization steps $T_{\text{adv}}$, access to loss function $g(X_{\text{forget}}^{\text{adv}}) :=$ $\mathcal{L}_{\text{retain}}(\mathsf{U}(f_{\text{target}}, \{X_{\text{forget}}^{\text{adv}}, y_{\text{forget}}\}, \mathcal{D}_{\text{retain}}))$, hyperparameters $p, m$.
2: Initialize $X_{\text{forget}}^{\text{adv}} \leftarrow X$ and initialize a random noise candidate set NoiseCandidates of size 1
3: **for** $t = 1, \cdots, T_{\text{adv}}$ **do**
4:     **for** $\mathbf{z} \in$ NoiseCandidates **do**
5:         **for** $i = 1, \ldots, p$ **do**
6:             $\mathbf{z}' \leftarrow$ ESTIMATEGRADIENTS$(\mathbf{z}, g)$         /* Call to gradient estimation procedure */
7:             Append $\mathbf{z}'$ to NoiseCandidates
8:         **end for**
9:     **end for**
10:     Keep the top $m$ noises in NoiseCandidates (based on loss function $g$)
11: **end for**
12: Choose the best $\mathbf{z}$ in NoiseCandidates
13: **Return:** $(X_{\text{forget}} + \mathbf{z}, y_{\text{forget}})$

14: **procedure** ESTIMATEGRADIENTS$(\mathbf{z}, g)$              /* Estimate the gradient and update noise */
15:     Draw random unit noise $\Delta$
16:     Compute $g(X_{\text{forget}} + \mathbf{z} + \Delta)$ and $g(X_{\text{forget}} + \mathbf{z} - \Delta)$
17:     **if** $g(X_{\text{forget}} + \mathbf{z} + \Delta) \leq g(X_{\text{forget}} + \mathbf{z})$ **and** $g(X_{\text{forget}} + \mathbf{z} - \Delta) \leq g(X_{\text{forget}} + \mathbf{z})$ **then**
18:         **Return:** $\emptyset$                  /* If no improvement, skip this estimator */
19:     **end if**
20:     Estimate gradient $\nabla g(\mathbf{z}) \leftarrow \frac{g(X_{\text{forget}} + \mathbf{z} + \Delta) - g(X_{\text{forget}} + \mathbf{z} - \Delta)}{2} \Delta$
21:     Update noise: $\mathbf{z}' \leftarrow \mathbf{z} + \eta_{\text{adv}} \nabla g(\mathbf{z})$
22:     **Return:** updated noise $\mathbf{z}'$
23: **end procedure**

---

# 3 EXPERIMENTS

We describe our experimental setup in § 3.1 and present results for white-box and black-box attacks in § 3.2 and § 3.3, respectively.

## 3.1 EXPERIMENTAL SETUP

**Datasets and models.** We evaluate the proposed attack on image classification tasks, one of the most common applications of machine unlearning (Bourtoule et al., 2021; Graves et al., 2021; Gupta et al., 2021; Chundawat et al., 2023; Tarun et al., 2023). We perform experiments across two testbeds:

- **CIFAR-10:** We use the model provided by the Machine Unlearning Challenge at NeurIPS 2023 (Triantafillou et al., 2023) as the target model $f_{\text{target}}$. This model is a ResNet-18 (He et al., 2016) trained on the CIFAR-10 dataset (Krizhevsky et al., 2009). We randomly select examples from the CIFAR-10 training set to form the forget set $\mathcal{D}_{\text{forget}}$, while the rest of the training data forms the retain set $\mathcal{D}_{\text{retain}}$. The CIFAR-10 test set is used as $\mathcal{D}_{\text{holdout}}$.
- **ImageNet:** We construct a larger-scale testbed using ImageNet. Here, we use a ResNeXt-50 model (Xie et al., 2017) pretrained on ImageNet (Deng et al., 2009) as the target model $f_{\text{target}}$. Similar to CIFAR-10, we randomly select examples from the ImageNet training set to create the forget set $\mathcal{D}_{\text{forget}}$, with the remaining data forms $\mathcal{D}_{\text{retain}}$. The ImageNet test set is used as $\mathcal{D}_{\text{holdout}}$.

**Unlearning algorithms.** Our evaluation focuses on *Gradient Ascent* (GA) and two of its variants, which perform unlearning by continuing to train on the forget set (and optionally the retain set). The main difference among these methods lies in their objective functions:

- Gradient Ascent (GA). GA maximizes the cross-entropy loss on the forget set $\mathcal{D}_{\text{forget}}$, denoted by $\mathcal{L}_{\text{forget}}$. GA is notably one of the most popular unlearning algorithms, as demonstrated by its use as a baseline in the Machine Unlearning Challenge at NeurIPS 2023 (Triantafillou et al., 2023).
- Gradient Ascent with Gradient Descent on the Retain Set (GA$_{\text{GDR}}$; Liu et al., 2022; Maini et al., 2024; Zhang et al., 2024). GA$_{\text{GDR}}$ minimizes the retain loss after removing the forget loss, denoted by $-\mathcal{L}_{\text{forget}} + \mathcal{L}_{\text{retain}}$, where $\mathcal{L}_{\text{retain}}$ is the cross-entropy of the retain set $\mathcal{D}_{\text{retain}}$. The motivation is to train the model to maintain its performance on $\mathcal{D}_{\text{retain}}$.
- Gradient Ascent with KL Divergence Minimization on the Retain Set (GA$_{\text{KLR}}$; Maini et al., 2024; Zhang et al., 2024). GA$_{\text{KLR}}$ encourages the unlearned model's probability distribution $p_{f_{\text{unlearn}}}(\cdot|x)$ to be close to the target model's distribution $p_{f_{\text{target}}}(\cdot|x)$ on inputs from the retain set $x \in \mathcal{D}_{\text{retain}}$. Specifically, the objective loss to minimize is $-\mathcal{L}_{\text{forget}} + \text{KL}_{\text{retain}}$, where $\text{KL}_{\text{retain}}$ is the KL Divergence between $p_{f_{\text{unlearn}}}(\cdot|x)$ and $p_{f_{\text{target}}}(\cdot|x)$ with $x$ from the retain set.

For all unlearning algorithms, we use SGD as the optimizer, with a momentum of 0.9 and a weight decay of $5 \times 10^{-4}$. The (un)learning rate is set to 0.02 for CIFAR-10 and 0.05 for ImageNet. Each unlearning process is run with a batch size of 128 for a single epoch.[3]

**Measuring attack performance.** We quantify attack effectiveness by measuring the accuracy degradation introduced by the adversarial forget set on the unlearned model. For a given forget set $\mathcal{D}_{\text{forget}}$, we create two models: (i) $f_{\text{unlearn}}$, by applying the unlearning algorithm to $\mathcal{D}_{\text{forget}}$, and (ii) $f_{\text{unlearn}}^{\text{adv}}$, by generating an adversarial forget set $\mathcal{D}_{\text{forget}}^{\text{adv}}$ of the same size using our attack, and applying the unlearning algorithm to $\mathcal{D}_{\text{forget}}^{\text{adv}}$. We quantify the attack's performance as the maximum accuracy degradation observed across a grid search of hyperparameters, defined as $\Delta \text{Acc}_{\mathcal{D}} := \max_{\lambda} \left( \text{Acc}(f_{\text{unlearn}}; \mathcal{D}) - \text{Acc}(f_{\text{unlearn}}^{\text{adv}}; \mathcal{D}) \right)$, where $\lambda$ denotes the attack's hyperparameters (detailed in § A), and $\mathcal{D}$ represents the evaluation dataset. In our evaluation, for general attacks, we report the accuracy drop on the retain and holdout sets, referred to as $\Delta \text{Acc}_{\text{retain}}$ and $\Delta \text{Acc}_{\text{holdout}}$. For targeted attacks, we report the accuracy drop on a specific set of examples.

We vary the size of the forget set, $|\mathcal{D}_{\text{forget}}|$, from 10 to 100 in our evaluation. To ensure the statistical significance of our results, for each evaluated $|\mathcal{D}_{\text{forget}}|$, we randomly select five different subsets for $\mathcal{D}_{\text{forget}}$ and report both the maximum and mean measurements. The runtime of our attack is provided in § A.

---

[3]Note that if the forget set size is smaller than 128, this would be equivalent to a single-step unlearning.

Table 1: Accuracy degradation of the unlearned model under white-box attacks on $\mathcal{D}_{\text{retain}}$ and $\mathcal{D}_{\text{holdout}}$ across varying sizes of forget set $\mathcal{D}_{\text{forget}}$. For each size, 5 random forget sets were sampled, and hyperparameter search was conducted to optimize attack performance. The table reports the maximum, mean, and standard deviation of the accuracy drop across these sets. Consistent and significant accuracy degradation is observed.

| $|\mathcal{D}_{\text{forget}}|$ | $\Delta\text{Acc}_{\text{retain}}$: Accuracy drop on $\mathcal{D}_{\text{retain}}$ (%) | | | $\Delta\text{Acc}_{\text{holdout}}$: Accuracy drop on $\mathcal{D}_{\text{holdout}}$ (%) | | |
|---|---|---|---|---|---|---|
| | Max | Mean | Std | Max | Mean | Std |
| **CIFAR-10** | | | | | | |
| 10 | 93.73 | 90.68 | 2.76 | 82.26 | 80.20 | 2.63 |
| 20 | 95.12 | 93.23 | 1.13 | 84.66 | 82.29 | 1.31 |
| 50 | 94.37 | 91.92 | 1.25 | 83.26 | 80.84 | 1.34 |
| 100 | 95.45 | 92.39 | 1.65 | 84.40 | 81.47 | 1.67 |
| **ImageNet** | | | | | | |
| 10 | 70.56 | 31.98 | 26.79 | 78.40 | 34.57 | 35.16 |
| 20 | 96.37 | 79.63 | 21.25 | 86.88 | 69.32 | 21.66 |
| 50 | 96.81 | 94.79 | 2.07 | 88.64 | 86.23 | 2.25 |
| 100 | 96.88 | 96.68 | 0.26 | 88.68 | 88.30 | 0.52 |

## 3.2 WHITE-BOX ATTACK

We start with results for the white-box attack described in Algorithm 1.

**Accuracy degradation consistently reaches 90% across varying sizes of the forget set.** As shown in Table 1, the unlearned model experiences substantial accuracy degradation under the white-box attack. The maximum accuracy drop on the retain set $\mathcal{D}_{\text{retain}}$ consistently hits around 90% for both CIFAR-10 and ImageNet (§ B.2 details the attack's performance across different hyperparameters). However, when the forget set is smaller (e.g., $|\mathcal{D}_{\text{forget}}| \leq 10$), the variability in accuracy degradation increases across forget sets sampled with different random seeds. This variability likely arises from the unlearning algorithm's sensitivity when dealing with a small forget set. In other words, this reveals a new vulnerability where an attacker might experiment with different forget set sizes to identify the most effective one for compromising the unlearning pipeline.

For reference, Figure 2 visualizes the forget sets before and after the attack on CIFAR-10 and ImageNet. The changes in adversarial forget sets are visually minimal.

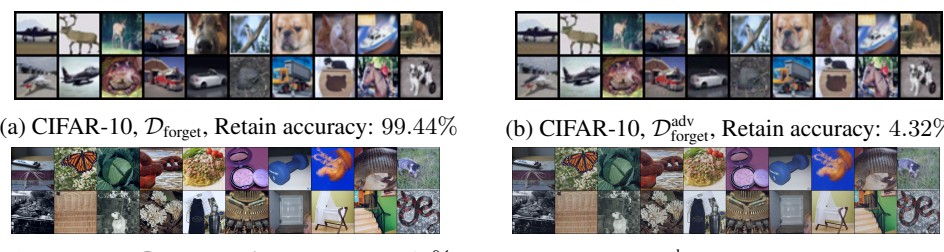

(a) CIFAR-10, $\mathcal{D}_{\text{forget}}$, Retain accuracy: 99.44%    (b) CIFAR-10, $\mathcal{D}_{\text{forget}}^{\text{adv}}$, Retain accuracy: 4.32%

(c) ImageNet, $\mathcal{D}_{\text{forget}}$, Retain accuracy: 96.41%    (d) ImageNet, $\mathcal{D}_{\text{forget}}^{\text{adv}}$, Retain accuracy: 0.04%

Figure 2: Visualization of the original forget set $\mathcal{D}_{\text{forget}}$ (a, c) and adversarial forget sets $\mathcal{D}_{\text{forget}}^{\text{adv}}$ (b, d) for CIFAR-10 and ImageNet. Although the adversarial forget sets appear visually similar to the original valid forget set, they lead to catastrophic accuracy failure in the unlearned model.

**Theoretical insights.** In § C, we provide theoretical insights into our empirical findings. Specifically, we demonstrate that in a stylized setting of unlearning for linear models, there exists an attack wherein the predictor returned by a GA-like unlearning algorithm applied on a small perturbation of $\mathcal{D}_{\text{forget}}$ misclassifies all examples in $\mathcal{D}_{\text{retain}}$.

**Targeted attacks are also highly effective.** We then craft an adversarial forget set to induce a targeted accuracy drop in the unlearned model by modifying our attack algorithm to compute the target loss using examples from the same class. We apply this attack to CIFAR-10, and report the resulting accuracy drop across different targeted classes in Figure 3 (with the accuracy drop on non-targeted classes kept below 10%). The attack consistently causes a significant accuracy degradation in targeted classes.

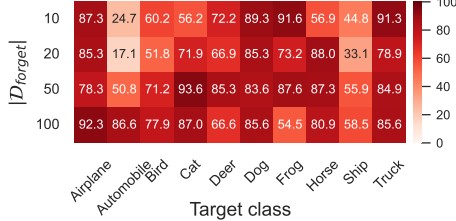

Figure 3: Accuracy drop (%) across different targeted classes in CIFAR-10 after targeted attack, with impact on non-targeted classes kept < 10%.

**Transferability across models.** We also explore the transferability of adversarial forget sets across different models. We consider two models:

- *Target model*: the CIFAR-10 model from the unlearning competition (Triantafillou et al., 2023), with training details unknown.
- *Shadow model*: a ResNet-18 model we train on CIFAR-10 using SGD for 50 epochs (learning rate 0.01, momentum 0.9).

Table 2: Accuracy drop (%) on retain set of CIFAR-10 using adversarial forget sets, found in the shadow model and applied to target model. Adversarial forget sets show transferability across models.

|      | **Max** | **Mean** | **Std** |
|------|---------|----------|---------|
| 10   | 68.04   | 48.36    | 13.48   |
| 20   | 37.41   | 29.38    | 7.93    |
| 50   | 14.24   | 11.63    | 3.39    |
| 100  | 3.91    | 3.63     | 0.30    |

We then generate the adversarial forget set on the shadow model and test its transferability to the target model. Table 2 shows that these sets still cause a significant accuracy drop in the target model. This transferability is more effective with smaller forget sets.

**Generalization to other unlearning methods.** We also evaluate the effectiveness of our white-box attack against other unlearning methods, specifically $GA_{GDR}$ and $GA_{KLR}$ (see §3.1). As shown in Table 3, our attack remains highly effective, even when the unlearning algorithm includes regularization terms: These terms are intended either to preserve the model's performance on the retained set by penalizing any increase in loss ($GA_{GDR}$) or to prevent the unlearned model from deviating too much from the original ($GA_{KLR}$).

Table 3: Maximum accuracy drop (%) with (of 5 randomly selected forget sets for each size) under white-box attacks on $\mathcal{D}_{retain}$. The attack is consistently effective across different unlearning methods.

| $|\mathcal{D}_{forget}|$ | **CIFAR-10** | | **ImageNet** | |
|--------------------------|--------------|--------------|--------------|--------------|
|                          | $GA_{GDR}$   | $GA_{KLR}$   | $GA_{GDR}$   | $GA_{KLR}$   |
| 10                       | 91.91        | 92.66        | 93.95        | 94.99        |
| 20                       | 93.55        | 94.01        | 93.69        | 93.71        |
| 50                       | 93.45        | 91.97        | 96.62        | 96.79        |
| 100                      | 89.26        | 92.79        | 96.69        | 96.67        |

## 3.3 BLACK-BOX ATTACK

In the black-box attack setting, the attacker only has query access to the model without knowing its weights. They must estimate the gradient of the retain loss after running the unlearning algorithm with adversarial examples to improve their generation of those examples (see Algorithm 2).

Despite this, as shown in Table 4, the attack remains highly effective, causing up to an $89.6\%$ drop in retain accuracy on CIFAR-10 and $80.3\%$ on ImageNet. However, we also observe that the black-box attack is generally more effective with smaller forget sets. This observation is likely consistent with previous research on zeroth-order optimization, demonstrating that its performance deteriorates in high-dimensional settings, as shown by previous works (e.g., Duchi et al. (2015)).

Table 4: Accuracy drop of the unlearned model under *black-box* attacks on $\mathcal{D}_{retain}$ and $\mathcal{D}_{holdout}$ across varying sizes of forget set $\mathcal{D}_{forget}$. For each size, 5 random forget sets were sampled, and hyperparameter search was conducted to optimize attack performance. The table reports the maximum, mean, and standard deviation of the accuracy drop across these sets.

| $|\mathcal{D}_{forget}|$ | $\Delta Acc_{retain}$: Accuracy drop on $\mathcal{D}_{retain}$ (%) | | | $\Delta Acc_{holdout}$: Accuracy drop on $\mathcal{D}_{holdout}$ (%) | | |
|--------------------------|------|------|------|------|------|------|
|                          | Max  | Mean | Std  | Max  | Mean | Std  |
| **CIFAR-10**             |      |      |      |      |      |      |
| 10                       | 42.06 | 38.97 | 4.36 | 37.96 | 35.60 | 3.34 |
| 20                       | 35.01 | 27.02 | 11.31 | 31.74 | 24.65 | 10.03 |
| 50                       | 20.32 | 15.59 | 6.68 | 17.74 | 14.20 | 5.01 |
| 100                      | 15.30 | 11.89 | 2.49 | 12.84 | 10.13 | 2.03 |
| **ImageNet**             |      |      |      |      |      |      |
| 10                       | 80.32 | 69.03 | 16.64 | 72.10 | 62.42 | 13.78 |
| 20                       | 92.83 | 59.54 | 42.86 | 84.42 | 53.82 | 38.41 |
| 50                       | 94.61 | 55.80 | 45.55 | 86.32 | 50.41 | 44.88 |
| 100                      | 56.04 | 26.43 | 29.55 | 49.72 | 23.68 | 25.88 |

## 4 EXPLORATION OF DEFENSIVE MECHANISMS

In this section, we explore potential defenses that model deployers could implement to detect adversarial unlearning requests. We consider two scenarios: (i) when the deployer cannot store all

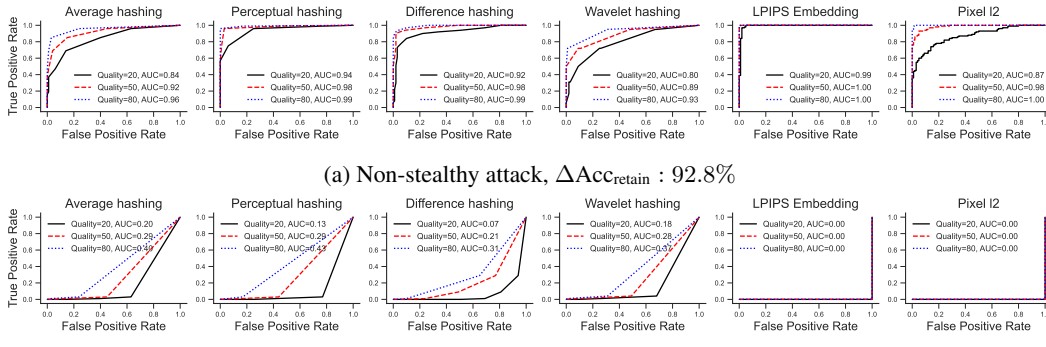

(a) Non-stealthy attack, $\Delta\text{Acc}_{\text{retain}}: 92.8\%$

(b) Stealthy attack (w/ projection norm = 10), $\Delta\text{Acc}_{\text{retain}}: 84.1\%$

Figure 4: ROC curves for various detection methods under (a) non-stealthy attacks and (b) stealthy attacks, where adversarial changes are projected onto a valid input space constrained by an $\ell_2$ ball. The stealthiness of adversarial forget sets makes them harder to differentiate from benign requests with natural perturbations, resulting in lower AUC.

training examples due to practical limitations such as storage constraints (§ 4.1), and (ii) when the deployer has access to all training examples (§ 4.2).

## 4.1 DEFENSES WITH LIMITED ACCESS TO TRAINING EXAMPLES

When storage constraints prevent retaining full copies of training data, a possible defense is to store hashes or embeddings of the original examples instead and compare them to incoming unlearning requests; a request is considered invalid if its distance from any stored hash or embedding is above a threshold $\tau$.

**Evaluation setup.** We evaluate the effectiveness of these detection mechanisms using a forget set consisting of 100 CIFAR examples. Adversarial examples are generated either without constraints or by applying an $\ell_2$ norm limit (via projection onto the $\ell_2$ ball during each adversarial optimization step) to enhance stealthiness. To ensure the detection mechanism can also accommodate benign user requests, which may involve slight variations (e.g., a re-encoded version of the original image), we simulate benign modifications by re-encoding the image data with a standard JPEG encoder. Specifically, for each image in the forget set, we use a saved copy of it using standard JPEG encoder[4] with different `quality` parameters[5]. A lower `quality` value generally results in more information loss, making the encoded image less similar to the original pixel values. Therefore, allowing benign requests becomes more challenging at lower `quality` levels, as the mechanism must tolerate greater deviations from the original images while maintaining accuracy in detection.

We evaluate four image hashing methods—average hashing, perceptual hashing, difference hashing, and wavelet hashing (Little, 2024)—as well as one image embedding method, LPIPS (Zhang et al., 2018). We report the AUROC across different distance thresholds $\tau$ for each method. Higher AUROC values indicate better detection of adversarial requests while maintaining tolerance for benign user perturbations.

**Stealthy attack can easily escape the detection.** As shown in Figure 4a, detection methods are fairly effective when adversarial modifications to the original images are unbounded, with AUROC values exceeding 0.7 in most cases. However, when the attacker increases stealth by limiting the adversarial modifications' $\ell_2$ norm (see Figure 4b), although this slightly reduces the attack's effectiveness, it allows the attack to evade most detection mechanisms, often causing the AUROC to drop well below 0.5, meaning the adversarial forget sets become more similar to the original forget set than naturally perturbed sets (i.e., those generated by image compression). Interestingly, while LPIPS is highly effective when stealth is not enabled, it becomes the weakest method once stealthy tactics are applied.

---

[4]We use the Python Imaging Library.

[5]Note JPEG is a lossy image codec—data loss could occur even with `quality=100`. Commonly used value ranges balancing storage and quality are 70–80. For example, libjpeg-turbo set the default `quality=75` for the command line tool example.

## 4.2 Defenses with Full Access to Training Examples

When deployers have access to the complete set of training examples, their defensive capabilities are significantly enhanced. We consider two approaches:

**Distance checking:** Similar to the detection methods discussed in § 4.1, the deployer can directly compute pixel-wise $\ell_2$ distances between incoming unlearning requests and stored training images. A request is considered invalid if its pixel distance from any stored image is above a threshold $\tau$. However, as shown in Figure 4, stealthy attacks can still evade this pixel-based checker.

**Similarity searching and indexing:** To address the limitations of simple distance checking in capturing adversarial requests, we propose an alternative unlearning protocol: Instead of directly accepting incoming images, the deployer performs a similarity search against stored training images and retrieves the closest matches for use in the unlearning process. This protocol prevents adversarial examples from directly entering the unlearning pipeline.

However, this protocol still has a potential vulnerability: attackers may optimize the selection of examples in the forget set to maximize negative impact on model performance. As shown in Figure 5, such attacks remain feasible on GA when the forget set is small. For instance, with a forget set containing only ten examples, an adversarially selected set could result in a retain error of 30.9%, while a normal forget set gives a retain error of 3.8%. In other words, even when the assumption $\mathcal{D}_{\text{forget}} \subset \mathcal{D}_{\text{train}}$ holds, attackers could still exploit other avenues to compromise the unlearning process. We also investigate this vulnerability for exact unlearning in § B.3 and find it becomes more sensitive when the forget set is large.

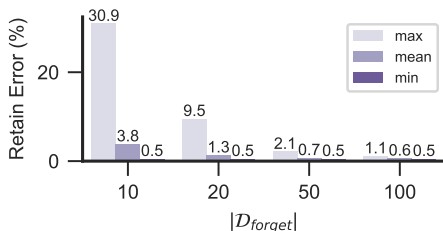

Figure 5: The attacker can optimize the selection of examples in a valid forget set to degrade performance. We report the maximum, mean, and minimum retain error on CIFAR-10 for varying $|\mathcal{D}_{\text{forget}}|$, with 3000 random selections per size.

## 5 Discussions

Finally, we discuss how relaxing the threat model described in § 2.1 could affect the attack's performance. In general, our findings below indicate that the attack can succeed even under weaker threat models. This versatility makes developing an effective defense even more challenging.

### 5.1 Successful attack even without knowing the exact unlearning algorithm

In § 2.1, we assume the attacker has the knowledge of the deployed unlearning algorithm. Here, we consider a more relaxed scenario where the attacker is unaware of the specific algorithm in use. Specifically, the attacker designs a malicious unlearning request for algorithm A (one of GA or $\text{GA}_{\text{GDR}}$ or $\text{GA}_{\text{KLR}}$), while the actual system implements algorithm B (also one of GA or $\text{GA}_{\text{GDR}}$ or $\text{GA}_{\text{KLR}}$). As shown in Table 5, even without knowledge of the exact unlearning algorithm, the attack still results in an accuracy drop of approximately 90% on the retain set.

Table 5: Transferability of adversarial forget sets across unlearning algorithms used to generate them (rows) and the actual unlearning algorithms (columns). Numbers in the table are the maximum accuracy drop (%) with $|\mathcal{D}_{\text{forget}}| = 20$ under white-box attacks on $\mathcal{D}_{\text{retain}}$. Diagonal entries represent cases where the attacker knows the exact algorithm.

|  | GA | $\text{GA}_{\text{GDR}}$ | $\text{GA}_{\text{KLR}}$ |
|---|---|---|---|
| **GA** | 95.12 | 93.52 | 93.75 |
| **$\text{GA}_{\text{GDR}}$** | 94.20 | 93.55 | 94.01 |
| **$\text{GA}_{\text{KLR}}$** | 93.52 | 91.30 | 93.07 |

### 5.2 Successful attack even without access to training dataset

Another assumption is that the attacker is a data owner within the system and has access to a collection of examples they previously submitted, which were used in training. We now explore the feasibility of the attack when the attacker is not part of the training process and, therefore, does not have access to any portion of the training dataset.

We randomly select images from CIFAR-100 (Krizhevsky et al., 2009) or even use randomly initialized pixels to launch the black-box attack. As shown in Table 6, even when the attacker initializes their adversarial forget sets with examples not present in the training set, the attack remains effective. However, we note that detection mechanisms discussed in § 4 would likely catch these attempts.

Table 6: Accuracy drop (%) on $\mathcal{D}_{\text{retain}}$ for CIFAR-10, with adversarial forget sets initialized from non-training examples.

| $|\mathcal{D}_{\text{forget}}|$ | Random pixels | CIFAR-100 |
|---|---|---|
| 10 | 67.14 | 30.69 |
| 20 | 65.31 | 26.63 |
| 50 | 44.16 | 14.46 |
| 100 | 20.89 | 8.63 |

## 6  RELATED WORK

**Machine unlearning.** Machine unlearning is an emerging research area motivated by privacy regulations like GDPR (European Parliament & Council of the European Union) and their ethical considerations, such as the right to be forgotten. It focuses on enabling models to "forget" specific training data, and has been studied in image classification, language models, and federated learning (Cao & Yang, 2015; Garg et al., 2020; Liu et al., 2020; Meng et al., 2022; Huang et al., 2022; Che et al., 2023; Cohen et al., 2023; Wu et al., 2023). Exact unlearning guarantees that the unlearned model is indistinguishable from the model retrained without the forget set. This is generally only feasible for simple models like SVMs and naive Bayes classifiers (Cauwenberghs & Poggio, 2000; Tveit et al., 2003; Romero et al., 2007; Karasuyama & Takeuchi, 2010). For more complex models such as neural networks, efficient exact unlearning remains an open question. Approximate unlearning offers a more practical solution here, aiming to erase the influence of the forget set without formal guarantees. For example, a widely used family of unlearning algorithms problems performs gradient ascent on the training loss computed on the forget data (Guo et al., 2020; Izzo et al., 2021).

**Side effects of machine unlearning.** Recent studies have raised concerns about various side effects of machine unlearning. Carlini et al. (2022) introduce the privacy onion effect, where removing the most vulnerable data points exposes deeper layers of previously safe data to attacks. Di et al. (2022) show that a camouflaged poisoning attack can be enabled via carefully designed unlearning requests. Hayes et al. (2024) highlight how inexact unlearning often overestimate privacy protection. Shi et al. (2024) show that unlearning methods might still exhibit partial memorization of unlearned data, providing a false sense of security regarding privacy guarantees. Shumailov et al. (2024) introduces "ununlearning" and shows how previously unlearned knowledge can be reintroduced through in-context learning. Łucki et al. (2024) demonstrate that recent unlearning methods proposed for model safety only superficially obscure harmful knowledge rather than fully removing it. Our work focus on a different potential vulnerability in the current machine unlearning protocol, showing that adversarially perturbed forget sets can severely degrade model performance. Furthermore, we show the challenge of efficiently and robustly verifying the authenticity of unlearning requests.

**Adversarial examples.** Szegedy (2013) show that neural networks could be easily misled by slightly perturbing the input images, causing models to make incorrect predictions with high confidence. This phenomenon has been extensively studied from both the attack and defense sides (e.g., Goodfellow et al., 2015; Moosavi-Dezfooli et al., 2017; Mądry et al., 2018; Kurakin et al., 2018). Our attack also adds adversarial noise to images but with the more challenging goal of fooling a future model produced by unlearning the adversarially perturbed examples.

## 7  CONCLUSION

In this work, we reveal a critical vulnerability in machine unlearning systems, where adversaries can significantly degrade model accuracy by submitting unlearning requests for data not in the training set. Our case study on GA and its variants, a family of widely used unlearning methods, shows that after the attack, test accuracy drops to $3.6\%$ on CIFAR-10 and $0.4\%$ on ImageNet (white-box), and $8.5\%$ on CIFAR-10 and $1.3\%$ on ImageNet (black-box). Moreover, even advanced verification mechanisms fail to detect these attacks without hindering legitimate requests, highlighting the need for stronger verification methods to secure unlearning systems.

We also recognize several limitations in our work and suggest directions for future research. First, we focus on image classification, and future research could explore the attack's transferability to unlearning for language models (Eldan & Russinovich, 2023; Zhang et al., 2024; Maini et al., 2024; Shi et al., 2024). Second, our work targets GA-based unlearning, and future research could examine the applicability of our findings to other approaches, including non-differentiable methods such as task vectors (Ilharco et al., 2023). Lastly, while we have explored various defense mechanisms, there may be additional strategies that warrant investigation for their potential robustness.

ACKNOWLEDGMENTS

We thank Matthew Jagielski, Eleni Triantafillou, Karolina Dziugaite, Ken Liu, Andreas Terzis, Weijia Shi, and Nicholas Carlini for their valuable feedback and discussions.

Part of this work was completed while Yangsibo Huang was a PhD student at Princeton, and she acknowledges the support of the Wallace Memorial Fellowship and a Princeton SEAS Innovation Grant.

ETHICS STATEMENT

Our work examines a critical vulnerability in machine unlearning systems, specifically how adversarial unlearning requests can potentially degrade model accuracy. A potential concern would be that an adversarial attacker could deploy our algorithm to attack an existing system with machine unlearning capabilities. However, we believe the risk at the current moment is low because while machine unlearning is being actively researched on as a promising paradigm, it has not yet been widely deployed. For future deployments, our work is actually important for improving the security. Specifically, by exploring these security risks, we aim to 1) disclose and highlight potential threats, and 2) advance understanding and call for improvements such as stronger verification mechanisms for making unlearning protocols safer and more robust.

All our experiments, including white-box and black-box attacks, were conducted strictly for academic purposes in order to better understand the security of machine unlearning methods. We have not engaged in any malicious activities, and the datasets used (CIFAR-10 and ImageNet) are publicly available, and widely used for research purposes.

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

# Appendices

## A  Experimental details

We provide more experimental details as follow.

### A.1  Compute Configuration

We conduct all the experiments on NVIDIA A100-64GB GPU cards with 4 CPUs. The typical average runtime for different experiments are listed in Table 7 with five random drawn forget datasets. For white-box attack, we report the average runtime with $T_{\text{adv}} = 1000$. For black-box attack, we report the runtime with $T_{\text{adv}} = 1000, m = 1, p = 1, d = 5$.

Table 7: Average runtime (in GPU seconds) for the attack across five independent runs, reported under different forget set sizes. The standard deviations are shown in parentheses.

| $|\mathcal{D}_{\text{forget}}|$ | White-box Attack | Black-box attack |
|---|---|---|
| 10 | 71.72 | 552.98 |
| 20 | 71.30 | 544.54 |
| 50 | 71.41 | 548.54 |
| 100 | 72.20 | 549.60 |

(a) CIFAR-10

| $|\mathcal{D}_{\text{forget}}|$ | White-box attack | Black-box attack |
|---|---|---|
| 10 | 808.27 | 944.86 |
| 20 | 794.51 | 1052.99 |
| 50 | 1095.13 | 1497.13 |
| 100 | 1758.28 | 2200.38 |

(b) ImageNet

### A.2  Hyperparameters

**Hyperparameters for unlearning.** The hyperparameters for unlearning are selected to ensure that the unlearning process on benign $\mathcal{D}_{\text{forget}}$ is effective, resulting in a roughly $10\%$ to $20\%$ drop in accuracy, closely aligning with the model's performance on non-training examples from $\mathcal{D}_{\text{forget}}$. At the same time, it induces only a minimal accuracy reduction on $\mathcal{D}_{\text{retain}}$.

**Hyperparameters for our attacks.** We also provide the hyperparameter settings for white-box and black-box attacks used in our experiments in Table 8.

Table 8: Hyperparameter settings for white-box and black-box attacks used in our experiments.

| | CIFAR-10 | ImageNet |
|---|---|---|
| White-box attack | | |
| Forget set size ($|\mathcal{D}_{\text{forget}}|$) | $\{10, 20, 50, 100\}$ | $\{10, 20, 50, 100\}$ |
| Adversarial step size ($\eta_{\text{adv}}$) | $\{0.01, 0.02, 0.05, 0.1, 0.2, 0.5, 1.0, 2.0, 5.0\}$ | |
| Adversarial optimization steps ($T_{\text{adv}}$) | $\{10, 20, 50, 100, 200, 500, 1000, 2000, 5000\}$ | |
| Black-box attack | | |
| Forget set size ($|\mathcal{D}_{\text{forget}}|$) | $\{10, 20, 50, 100\}$ | $\{10, 20, 50, 100\}$ |
| Adversarial step size ($\eta_{\text{adv}}$) | $\{0.01, 0.02, 0.05, 0.1, 0.2, 0.5, 1.0, 2.0, 5.0\}$ | |
| Adversarial optimization steps ($T_{\text{adv}}$) | $\{10, 20, 50, 100, 200, 500, 1000, 2000, 5000\}$ | |
| # of gradient estimators for each noise candidate ($p$) | $\{1, 2, 3, 4, 5\}$ | $\{1, 2, 3\}$ |
| Candidate noise set size ($m$) | $\{1, 2, 3\}$ | $\{1, 2, 3\}$ |

### A.3  Black-box Attack Algorithm for ImageNet

We make slight adjustments to the black-box algorithm for ImageNet, which we describe in Algorithm 3. The main differences between Algorithm 2 and Algorithm 3 are as follows:

- **Stopping condition:** in Algorithm 2, the update process for noise stops, and the loop proceeds to the next iteration if the conditions $g(X_{\text{forget}} + \mathbf{z} + \Delta) \leq g(X_{\text{forget}} + \mathbf{z})$ and $g(X_{\text{forget}} + \mathbf{z} - \Delta) \leq g(X_{\text{forget}} + \mathbf{z})$ are both satisfied. In contrast, we found this condition frequently occurs for ImageNet (Algorithm 3), so we allow the algorithm to continue without stopping, making the process more robust to such scenarios.

- **Gradient estimation parameter:** In Algorithm 3, we introduce an additional hyperparameter $d$ to control the number of gradient estimators. Specifically, $d$ gradient estimators are drawn and averaged to obtain the final gradient. This adjustment is made because white-box attacks tend to perform well on ImageNet, and averaging over a larger $d$ improves the quality of the gradient estimate.

---

**Algorithm 3** Black-box attack for ImageNet

---

1: **Input:** original model $f_{\text{target}}$, a collection of training examples $X \subset \mathcal{D}_{\text{train}}$, retain set $\mathcal{D}_{\text{retain}}$, unlearning method $\mathsf{U}$, adversarial step size $\eta_{\text{adv}}$, training steps $T_{\text{adv}}$, access to loss function $g(X_{\text{forget}}^{\text{adv}}) := \mathcal{L}_{\text{retain}}(\mathsf{U}(f_{\text{target}}, \{X_{\text{forget}}^{\text{adv}}, y_{\text{forget}}\}, \mathcal{D}_{\text{retain}}))$, hyperparameters $p, m, d$.
2: Initialize $X_{\text{forget}}^{\text{adv}} \leftarrow X$
3: Initialize noise candidate set $\text{NoiseCandidates}$ of size $m$
4: **for** $t = 1, \cdots, T_{\text{adv}}$ **do**
5:     **for** $\mathbf{z} \in \text{NoiseCandidates}$ **do**
6:         $\mathbf{z}' \leftarrow \text{EstimateGradientsImagenet}(\mathbf{z}, p, d, g)$     /* Call to the new ImageNet gradient estimation procedure */
7:         Append $\mathbf{z}'$ to $\text{NoiseCandidates}$
8:     **end for**
9:     Keep the top $m$ noises in $\text{NoiseCandidates}$ (based on loss function $g$)
10: **end for**
11: Choose the best $\mathbf{z}$ in $\text{NoiseCandidates}$
12: **Return:** $(X_{\text{forget}} + \mathbf{z}, y_{\text{forget}})$

13: **procedure** ESTIMATEGRADIENTSIMAGENET($\mathbf{z}, p, d, g$)    /* Estimate the gradient and update noise for ImageNet */
14:     **for** $i = 1, \cdots, p$ **do**                               /* Repeat $p$ times for noise candidates */
15:         **for** $j = 1, \cdots, d$ **do**  /* draw $d$ samples and compute the average for higher accuracy */
16:             Draw random unit noise $\Delta$
17:             Compute $g(X_{\text{forget}} + \mathbf{z} + \Delta)$ and $g(X_{\text{forget}} + \mathbf{z} - \Delta)$
18:             Estimate gradient $\nabla g_j(\mathbf{z}) \leftarrow \frac{g(X_{\text{forget}}+\mathbf{z}+\Delta)-g(X_{\text{forget}}+\mathbf{z}-\Delta)}{2}\Delta$
19:         **end for**
20:         Compute the average gradient estimator $\nabla g(\mathbf{z}) = \frac{1}{d}\sum_{j=1}^{d} \nabla g_j(\mathbf{z})$
21:         Update noise: $\mathbf{z}' \leftarrow \mathbf{z} + \eta_{\text{adv}} \nabla g(\mathbf{z})$
22:     **end for**
23:     **Return:** updated noise $\mathbf{z}'$
24: **end procedure**

---

# B   MORE EXPERIMENTAL RESULTS

## B.1   BENIGN UNLEARNING RESULTS

Table 9 reports the accuracy for the forget, retain, and test sets under benign unlearning requests of various sizes. As shown, the accuracy on the forget set is roughly comparable to the test set after unlearning, while the accuracy on the retain set remains high.

Table 9: Accuracy on forget, retain and test sets under benign unlearning request.

| $|\mathcal{D}_{\text{forget}}|$ | $f_{\text{unlearn}}$, Forget accuracy | $f_{\text{unlearn}}$, Retain accuracy | $f_{\text{unlearn}}$, Test accuracy |
|---|---|---|---|
| 10 | $83.15 \pm 14.82$ | $97.93 \pm 4.78$ | $86.88 \pm 3.92$ |
| 20 | $89.01 \pm 7.76$ | $99.09 \pm 1.04$ | $87.89 \pm 0.95$ |
| 50 | $86.18 \pm 3.74$ | $99.21 \pm 2.85$ | $88.03 \pm 0.24$ |
| 100 | $86.72 \pm 2.05$ | $99.36 \pm 6.18$ | $88.29 \pm 0.17$ |

## B.2   EFFECT OF ATTACK HYPERPARAMETERS

We further examine the performance of the white-box attack under different hyperparameter configurations. Results shown in Figure 6 reveal a clear trend: lower attack step sizes ($\eta_{\text{adv}}$) are more effective for attacks targeting smaller forget sets, while higher attack step sizes and increased optimization steps ($T_{\text{adv}}$) improve attacks on larger forget sets.

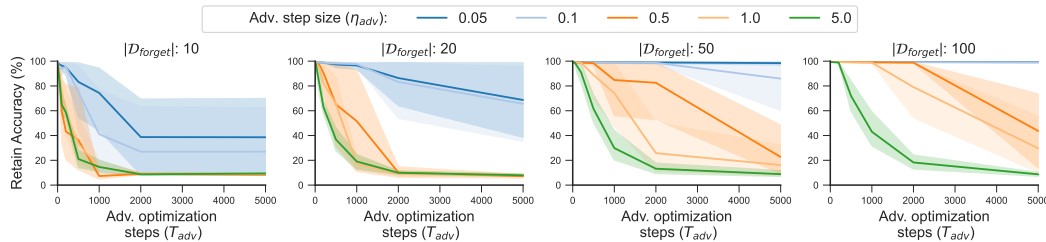

Figure 6: Retain accuracy after white-box attack on CIFAR-10 for various combinations of $T_{\text{adv}}$ and $\eta_{\text{adv}}$. Attacking a larger $\mathcal{D}_{\text{forget}}$ typically requires a higher $\eta_{\text{adv}}$ and larger $T_{\text{adv}}$.

## B.3   INDEXING ATTACK FOR EXACT UNLEARNING

Earlier in § 4.2, we discuss another potential protocol for unlearning: instead of directly accepting incoming images, the deployer performs a similarity search against stored training images and retrieves the closest matches for use in the unlearning process. However, we show that this protocol is still vulnerable in a sense that attackers may optimize the selection of examples in the forget set to maximize negative impact on model performance. For GA, the attack is most effective when $\mathcal{D}_{\text{forget}}$ is small.

We also investigate this vulnerability for exact unlearning in Figure 7 and find that its vulnerability slightly increases with a larger forget set. This is because, with a larger forget set, exact unlearning excludes more data points, making it slightly easier for certain selections of the forget set to cause a more significant accuracy drop.

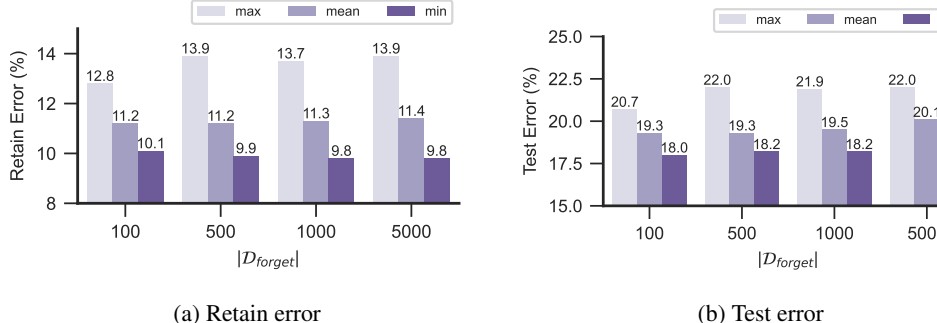

(a) Retain error                    (b) Test error

Figure 7: The attacker can optimize the selection of examples in a valid forget set to degrade performance. We report the maximum, mean, and minimum retain error on CIFAR-10 with exact unlearning for varying $|\mathcal{D}_{\text{forget}}|$, with 500 random selections per size.

## C    THEORETICAL DEMONSTRATION OF ATTACK

In this section, we discuss a simple stylized setting of a learning algorithm, an unlearning algorithm (that achieves perfect unlearning), and formally show that there exist adversarial perturbations of the forget set that can make the unlearning algorithm yield a predictor that performs poorly on the retain set.

### C.1    THE EXISTENCE RESULTS

Our goal is to construct a simple setting where it is easy to conceptually understand why such an attack can exist. We use an unlearning algorithm similar to the GA-family algorithms studied in our experiments, but we acknowledge a number of assumptions such as Gaussian inputs and linear models do not fully match the settings of our empirical results. Nonetheless, the formal results provide theoretical intuition on why such attacks are possible.

We discuss possible future directions to study more such constructions in § C.2.

For data $(x, y)$ drawn i.i.d. from a distribution $P$, a learning algorithm, abstractly speaking, is a (randomized) method $\mathsf{L} : (\mathcal{X} \times \mathcal{Y})^* \to \mathcal{H}$ that maps sequences $\mathcal{D}_{\text{train}} = ((x_1, y_1), \ldots, (x_n, y_n))$ of examples to a predictor $h \in \mathcal{H}$. We measure the quality of the learning algorithm in terms of its *empirical* and *population* loss, namely for $\ell : \mathcal{H} \times (\mathcal{X} \times \mathcal{Y}) \to \{0, 1\}$, we would like to minimize the population loss $\mathcal{L}(h; P) := \mathbb{E}_{(x,y) \sim P} \ell(h; x, y)$ or even the empirical loss $\mathcal{L}(h; \mathcal{D}_{\text{train}}) := \frac{1}{|\mathcal{D}_{\text{train}}|} \sum_{(x,y) \in \mathcal{D}_{\text{train}}} \ell(h; x, y)$.

An unlearning algorithm is a (randomized) method $\mathsf{U} : \mathcal{H} \times (\mathcal{X} \times \mathcal{Y})^* \times (\mathcal{X} \times \mathcal{Y})^* \to \mathcal{H}$ that maps the current predictor $h \in \mathcal{H}$, a *forget* set $\mathcal{D}_{\text{forget}} \in (\mathcal{X} \times \mathcal{Y})^*$ and a *retain* set $\mathcal{D}_{\text{retain}} \in (\mathcal{X} \times \mathcal{Y})^*$ to an updated predictor $\tilde{h} \in \mathcal{H}$. A desirable property of the unlearning algorithm $\mathsf{U}$ is that when invoked on $h$ being the output of the learning algorithm on input $\mathcal{D}_{\text{train}} = \mathcal{D}_{\text{forget}} \circ \mathcal{D}_{\text{retain}}$, $\mathsf{U}(\mathsf{L}(\mathcal{D}_{\text{forget}} \circ \mathcal{D}_{\text{retain}}), \mathcal{D}_{\text{forget}}, \mathcal{D}_{\text{retain}})$ has the same distribution as, or at least is "close to", the distribution $\mathsf{L}(\mathcal{D}_{\text{retain}})$; we say that $(\mathsf{L}, \mathsf{U})$ is a perfect *learning-unlearning* pair if this holds. A trivial way to achieve perfect learning-unlearning is by setting $\mathsf{U}(\mathsf{L}(\mathcal{D}_{\text{forget}} \circ \mathcal{D}_{\text{retain}}), \mathcal{D}_{\text{forget}}, \mathcal{D}_{\text{retain}}) = \mathsf{L}(\mathcal{D}_{\text{retain}})$. However, this method of unlearning is not desirable as it can be computationally inefficient to perform learning on $\mathcal{D}_{\text{retain}}$ from scratch. Instead, it is desired that $\mathsf{U}(\mathsf{L}(\mathcal{D}_{\text{forget}} \circ \mathcal{D}_{\text{retain}}), \mathcal{D}_{\text{forget}}, \mathcal{D}_{\text{retain}})$ has a computational complexity that is significantly less compared to that of $\mathsf{L}(\mathcal{D}_{\text{retain}})$.

We will now consider a specific realization of the learning and unlearning algorithms as follows.

**Data Distribution.** We consider the data distribution $P_{h^*}$ defined over $\mathbb{R}^d \times \{-1, 1\}$, parameterized by $h^* \in \mathbb{R}^d$, obtained by sampling $x \sim \mathcal{N}(0, \frac{1}{d} I_d)$ and setting $y = \text{sign}(\langle h^*, x \rangle)$.

We will consider the high dimensional regime where $n \ll d$.

**Loss Function.** We use the 0-1 loss function for halfspaces, $\ell(h; x, y) := \mathbb{1}\{y \neq \text{sign}(\langle h, x \rangle)\}$.

**Learning Algorithm.** We consider a *perceptron-like*[6] learning algorithm, defined as $\mathsf{L}(\mathcal{D}_{\text{train}}) := \sum_{(x,y) \in \mathcal{D}_{\text{train}}} y_i \cdot x_i$.

**Unlearning.** We consider the unlearning algorithm $\mathsf{U}(\hat{h}, \mathcal{D}_{\text{forget}}, \mathcal{D}_{\text{retain}}) := \hat{h} - \sum_{(x,y) \in \mathcal{D}_{\text{forget}}} y \cdot x$. Observe that in fact, the algorithm does not even use $\mathcal{D}_{\text{retain}}$, and thus is more efficient to compute than computing $\mathsf{L}(\mathcal{D}_{\text{retain}})$ from scratch.

**$\varepsilon$-Perturbation.** We say that $\mathcal{D}' = ((x_1', y_1), \ldots, (x_n', y_n))$ is an *$\varepsilon$-perturbation* of $\mathcal{D} = ((x_1, y_1), \ldots, (x_n, y_n))$ if for all $i$, it holds that $\|x_i' - x_i\|_2 \leq \varepsilon$.

We now present the main result of this section that shows the existence of an $\varepsilon$-perturbation of the forget set that makes the unlearning algorithm return a predictor that misclassifies all examples in the retain set.

**Theorem C.1** (Adversarial Forget Sets). *For all $\varepsilon, \beta > 0$ and $n < \sqrt{d/\log(d/\beta)}$ there exists $m = O(\sqrt{n}/\varepsilon)$ such that for $\mathcal{D}_{\text{train}}$ sampled i.i.d. from $P_{h*}$ with $|\mathcal{D}_{\text{train}}| = n$, $\mathcal{D}_{\text{forget}}$ being a randomly chosen subset of $\mathcal{D}_{\text{train}}$ with $|\mathcal{D}_{\text{forget}}| = m$ and $\mathcal{D}_{\text{retain}} := \mathcal{D}_{\text{train}} \setminus \mathcal{D}_{\text{forget}}$, all of the following hold with probability $1 - \beta$ over randomness of sampling the data:*

1. *$\mathcal{L}(\mathsf{L}(\mathcal{D}_{\text{train}}), \mathcal{D}_{\text{train}}) = 0$. In other words, the learned predictor achieves perfect accuracy on the training dataset.*
2. *$\mathsf{U}(\mathsf{L}(\mathcal{D}_{\text{forget}} \circ \mathcal{D}_{\text{retain}}), \mathcal{D}_{\text{forget}}, \mathcal{D}_{\text{retain}}) = \mathsf{L}(\mathcal{D}_{\text{retain}})$. In other words, $(\mathsf{L}, \mathsf{U})$ forms a perfect learning-unlearning pair.*
3. *There exists an $\varepsilon$-perturbation $\mathcal{D}'_{\text{forget}}$ of $\mathcal{D}_{\text{forget}}$ such that $\mathcal{L}(h'; \mathcal{D}_{\text{retain}}) = 1$ for $h' := \mathsf{U}(\mathsf{L}(\mathcal{D}_{\text{train}}), \mathcal{D}'_{\text{forget}}, \mathcal{D}_{\text{retain}})$. In other words, when provided a perturbation of $\mathcal{D}_{\text{forget}}$, the predictor after unlearning misclassifies the entire retain set $\mathcal{D}_{\text{retain}}$.*

We rely on the following concentration bound on the norms and inner products of random Gaussian vectors, which follow from simple applications of Bernstein's inequality for sub-exponential random variables.

**Fact C.2** (See, e.g., Vershynin (2018)). *For $x, y \sim \mathcal{N}(0, \sigma^2 I_d)$, it holds that*

$$\Pr[|\|x\|_2^2 - d\sigma^2| > \varepsilon d\sigma^2] \leq 2\exp\left(-\varepsilon^2 d/4\right),$$
$$\Pr[|\langle x, x' \rangle| > \varepsilon d\sigma^2] \leq 2\exp\left(-\varepsilon^2 d/4\right).$$

*Proof of Theorem C.1.* We first note that for $x_1, \ldots, x_n \sim \mathcal{N}(0, \frac{1}{d} I_d)$, we have from Fact C.2 that with probability $1 - \beta$, all of the following hold:

$$\|x_i\|_2^2 \geq 1 - \sqrt{\frac{\log(n/\beta)}{d}}, \tag{1}$$

$$\|x_i\|_2^2 \leq 1 + \sqrt{\frac{\log(n/\beta)}{d}}, \tag{2}$$

$$|\langle x_i, x_j \rangle| \leq 3\sqrt{\frac{\log(n/\beta)}{d}}. \tag{3}$$

We now proceed to prove each of the claimed parts, by conditioning on Eqs. (1), (2), and (3) holding. We use the notation $\mathcal{D}_{\text{train}} = ((x_1, y_1), \ldots, (x_n, y_n))$.

1. We have $\mathsf{L}(\mathcal{D}_{\text{train}}) = \hat{h} := \sum_i y_i \cdot x_i$. From Eqs. (1) and (3), we have that with probability $1 - \beta$, it holds for all $(x, y) \in \mathcal{D}_{\text{train}}$, that

$$y \cdot \langle \hat{h}, x \rangle = \|x\|_2 + \sum_{(x', y') \in \mathcal{D}_{\text{train}} \setminus \{(x,y)\}} yy' \cdot \langle x, x' \rangle$$
$$\geq 1 - \sqrt{\frac{\log(n/\beta)}{d}} - 3(n-1)\sqrt{\frac{\log(n/\beta)}{d}} > 0,$$

where we use that $n < O(\sqrt{d/\log(d/\beta)})$. Thus, with probability $1 - \beta$, we have that all examples in $\mathcal{D}_{\text{train}}$ are correctly classified by $\hat{h}$.

---

[6]The *Perceptron* algorithm (Rosenblatt, 1958) operates on the examples one at a time, by choosing an example $(x_i, y_i)$ that is misclassified by the current predictor $h$, and adds $y_i x_i$ to $h$. Here, we instead perform a single step on all examples at once.

As an aside, we note that under Eqs. (2) and (3), it also holds that

$$\begin{aligned}
\|\hat{h}\|^2 &= \sum_i \|x_i\|^2 + \sum_{i \neq j} y_i y_j \cdot \langle x_i, x_j \rangle \\
&\leq n \cdot \left(1 + \sqrt{\frac{\log(n/\beta)}{d}}\right) + \frac{n(n-1)}{2} \cdot \sqrt{\frac{\log(n/\beta)}{d}} \leq O(n).
\end{aligned}$$

2. This is immediate since $\mathsf{U}(\mathsf{L}(\mathcal{D}_{\text{train}}), \mathcal{D}_{\text{forget}}, \mathcal{D}_{\text{retain}}) = \sum_{(x,y) \in \mathcal{D}_{\text{train}}} y \cdot x - \sum_{(x,y) \in \mathcal{D}_{\text{forget}}} y \cdot x = \sum_{(x,y) \in \mathcal{D}_{\text{retain}}} y \cdot x$, which is precisely $\mathsf{L}(\mathcal{D}_{\text{retain}})$.

3. We consider the following $\varepsilon$-perturbation $\mathcal{D}'_{\text{forget}}$ of $\mathcal{D}_{\text{forget}}$, obtained by including $(x' = x + \varepsilon y \cdot \frac{\hat{h}}{\|\hat{h}\|}, y)$ in $\mathcal{D}'_{\text{forget}}$ for all $(x, y) \in \mathcal{D}_{\text{forget}}$. Note that this requires only knowledge of $\hat{h}$ and the forget set $\mathcal{D}_{\text{forget}}$; we do not need any information about the examples in $\mathcal{D}_{\text{retain}}$.

   The unlearning algorithm on this perturbed set would return: $\tilde{h} = \hat{h} - \sum_{(x,y) \in \mathcal{D}_{\text{forget}}} y \cdot x - \frac{\varepsilon m}{\|\hat{h}\|_2} \hat{h}$.

   Thus, for a suitable $m = O(\sqrt{n}/\varepsilon)$, we then have that $\tilde{h} = c\hat{h}$ for $c < 0$, and hence $y_i \cdot \langle \tilde{h}, x_i \rangle < 0$ for all $(x_i, y_i) \in \mathcal{D}_{\text{train}}$, and in particular, the returned predictor $\tilde{h}$ will misclassify the entire retain set. $\qquad\square$

## C.2 DISCUSSION

In summary, Theorem C.1 shows that for the learning-unlearning pair considered, small perturbations on a forget set that is much smaller than the training set size can cause the unlearning algorithm to return a predictor that misclassifies the entire training set. However, we note the following limitations of Theorem C.1, that we leave for future work to address.

First and foremost, the theorem is about a linear model, which does not capture the complex non-linearity of a neural network. Thus, it does not immediately provide any specific insight into the experiments we perform on neural networks in this paper.

Another limitation, is that we do not show that the learned predictor achieves small population loss. This is in fact impossible in the regime when $n \ll d$ as we consider. It would more desirable to have a learning-unlearning setting where the learning algorithm also achieves small population loss. However, we do note that the learning algorithm we consider is still "reasonable" in the sense that it does have generalization guarantees with $n \gg d$, on the family of distributions $\{P_{h^*} : h^* \in \mathbb{R}^d\}$ that we consider.

We suspect that the reason why the phenomenon occurs in neural networks is that small perturbations in feature (e.g., pixels) space can cause larger perturbations to the corresponding gradients. Whereas, in the setting of Theorem C.1, the attack arguably arises because the retain set examples are quite diverse from each other, and so a small forget set of size only $O(\sqrt{n}/\varepsilon)$ can perform a "coordinated attack" to push the predictor in a bad region.

Finally, another gap between our theoretical example and our experiments is that in the latter, we did not need a large forget set to execute an attack. In particular, the unlearning algorithm used *averaged* gradients over the forget set, so the attack could not have benefited just from the forget set being large.

