# OpenReview forum: "Unlearn and Burn: Adversarial Machine Unlearning Requests Destroy Model Accuracy"
_ICLR.cc/2025/Conference — ICLR 2025 Poster_

### Official Review · Reviewer_hRNb · 2024-10-31

**Soundness:** 3
**Presentation:** 3
**Contribution:** 2
**Rating:** 3
**Confidence:** 5

**Summary:**

This paper introduces Unlearn and Burn, an adversarial attack targeting machine unlearning algorithms to significantly degrade model performance by submitting malicious unlearning requests. The attack exploits the assumption that all unlearning requests correspond to legitimate training data, allowing an adversary to request the removal of data not present in the training set. The authors propose both white-box and black-box versions of the attack and demonstrate its effectiveness on CIFAR-10 and ImageNet, with results showing considerable drops in accuracy. The study also highlights the challenges of detecting such stealthy attacks, suggesting that existing verification methods for unlearning requests may be inadequate.

**Strengths:**

The paper presents an interesting and relevant idea, yet it has some notable issues that need addressing.

**Weaknesses:**

Exceeds page limit:  The paper currently extends beyond the 10-page maximum allowed by ICLR, utilizing in total 11 pages of core content. Additionally, several crucial aspects, such as unlearning algorithm specifics, hyperparameters, and theoretical results, are deferred to the appendix, making it challenging to assess the completeness of the methods presented within the main body.


Mischaracterization as a novel adversarial attack:  While the approach is framed as a new type of adversarial attack, it is more accurately an adaptation of established poisoning techniques. The attack uses a standard poisoning approach where the unlearning update replaces fine-tuning or training in the model. As outlined, the objective of minimizing model accuracy while removing poisoning samples mirrors conventional poisoning attacks. This is detailed in prior work such as [1-4] or survey [5] which already connect poisoning attacks to meta-learning approaches, similar to the attack formulation in Section 2.2 of this paper. Given these similarities, I observe a lacks of novelty compared to the state of the art and risk for creating confusion with respect to something that already exists in the literature.



Lack of novelty in attack mechanisms: The two attack methods (white-box and black-box) do not introduce any new methodologies beyond existing techniques. The white-box method is essentially a re-implementation of approaches discussed in [1] and [3], while the black-box method, based on gradient estimation, has been used in [6]. Consequently, the attack does not present a unique contribution in terms of adversarial methods.


Given the issues above, particularly the excessive page count and the lack of novelty, I recommend rejection of the paper. I encourage the authors to reconsider the framing of their approach, positioning it as an adaptation of poisoning attacks within unlearning settings rather than a new adversarial attack. I hope these comments will serve as constructive feedback for further refinement.



[1] Metapoison: Practical general-purpose clean-label data poisoning. NeurIPS 2020.

[2] The hammer and the nut: Is bilevel optimization really needed to poison linear classifiers? IJCNN 2021.

[3] Towards poisoning of deep learning algorithms with back-gradient optimization. ACM workshop on artificial intelligence and security 2017.

[4] Witches' brew: Industrial scale data poisoning via gradient matching ICLR 2021.

[5] Wild patterns reloaded: A survey of machine learning security against training data poisoning. ACM Computing Surveys 2024.

[6] Generative poisoning attack method against neural networks. arXiv 2017.

**Questions:**

Could you clarify the specific differences between your proposed attacks and prior poisoning methods?

**Details Of Ethics Concerns:**

The paper initially exceeded the 10-page maximum (by 2-3 lines) allowed by ICLR, totaling 11 pages of core content. ICLR explicitly states, “*The main text must be between 6 and 10 pages (inclusive).* ***This limit will be strictly enforced.***” Despite this, the paper has not been desk-rejected. I wanted to highlight this point to ensure that submission guidelines are consistently and fairly applied to all submissions.

---

> ### Author Response · Authors · 2024-11-21
> **Response to Reviewer hRNb**
>
> Thank you for your feedback! We're glad to hear that you find our paper interesting and the soundness and presentation to be good. We address your questions and comments below.
>
> **Q1: Exceeds page limit**
>
> > The paper currently extends beyond the 10-page maximum allowed by ICLR, utilizing in total 11 pages of core content.
> **A**: Thank you for pointing this out and we sincerely apologize for exceeding the page limit by 2 additional newlines, due to a last minute update that we did not carefully check.  We’ve fixed this in the revised PDF.
>
> > Additionally, several crucial aspects, such as unlearning algorithm specifics, hyperparameters, and theoretical results, are deferred to the appendix, making it challenging to assess the completeness of the methods presented within the main body.
>
> Thank you for the comment. Regarding the placement of technical content:
> - **Unlearning algorithm specifics, hyperparameters**: the core unlearning algorithms and their hyperparameters are fully described in the main text (lines 222-239). Only the hyperparameter selection rationale is in the appendix. We can certainly move this discussion to the main paper if you feel it would improve readability, space permitting.
> - **Theoretical result**: we have summarized the main theorem statement in the body of the paper (lines 298-301). This theorem establishes the existence of our proposed attacks in the context of linear model unlearning. The detailed proof remains in the appendix to maintain narrative flow while preserving technical completeness.
>
> **Q2 Mischaracterization as a novel adversarial attack & Lack of novelty in attack mechanisms**
>
> > While the approach is framed as a new type of adversarial attack, it is more accurately an adaptation of established poisoning techniques. The attack uses a standard poisoning approach where the unlearning update replaces fine-tuning or training in the model. As outlined, the objective of minimizing model accuracy while removing poisoning samples mirrors conventional poisoning attacks. This is detailed in prior work such as [1-4] or survey [5] which already connect poisoning attacks to meta-learning approaches, similar to the attack formulation in Section 2.2 of this paper.
>
> > The two attack methods (white-box and black-box) do not introduce any new methodologies beyond existing techniques. The white-box method is essentially a re-implementation of approaches discussed in [1] and [3], while the black-box method, based on gradient estimation, has been used in [5]. Consequently, the attack does not present a unique contribution in terms of adversarial methods.
>
> **A**:  We appreciate the reviewer’s comments and agree that our attacking algorithm is similar to some previous poisoning techniques. However, we note they all boil down to the same standard formulation with gradient based searching of adversarial inputs. The main novelty (of previous poisoning papers and our work) is applying such techniques to a novel application scenario and systematically studying the implications and defense. Specifically, we highlight that
> - Our primary contribution lies in identifying a novel **attack interface** specific to machine unlearning pipelines, which fundamentally targets a different system and interface (unlearning) compared to poisoning attacks (in standard learning or meta-learning).
> - With this new attack surface, we also explore corresponding defensive methods. We also note that the defensive landscape for unlearning differs significantly from that of training-time poisoning attacks. In unlearning, the unlearned set must resemble valid training data previously used to train the model. This enables defenses based on similarity comparison to training examples (Section 4).
>
> We discussed some of the previous poisoning attacks in the manuscript, but we apologize for missing some of the relevant citations and have addressed this in the revised manuscript.

---

> > ### Author Response · Authors · 2024-11-25
> > **Follow-up**
> >
> > Dear Reviewer,
> >
> > Thanks again for your detailed comments and suggestions. Since the rebuttal period is approaching its end, we would appreciate learning whether our responses have addressed your concerns. We are also happy to engage in further discussions.

---

> > > ### Comment · Reviewer_hRNb · 2024-11-27
> > > **Response**
> > >
> > > Thank you for your response and for revising the paper to comply with the conference guidelines. However, exceeding the 10-page limit could be considered unfair to other authors who followed the guidelines. ICLR explicitly states, "*The main text must be between 6 and 10 pages (inclusive).* ***This limit will be strictly enforced.***". Despite that, the paper has not been desk-rejected. Honestly, I can understand how such an oversight might occur, and I would feel personally disappointed if it had happened to me. I hope the authors can appreciate that my concern is rooted in a matter of fairness and is not intended as an outright criticism of the paper itself.
> > >
> > >
> > > Regarding the contribution, I appreciate the clarification provided by the authors. While the attack methods are adaptations of established poisoning techniques, I agree that exploring their application in the machine unlearning context is a valuable contribution. However, I see this value as being more *applicative* than *methodological* for the machine learning research community. That said, it may hold greater relevance and value for the security community, where its implications could be further explored and appreciated.
> > > With these considerations in mind, I have updated my score to reflect the significance of this application while maintaining my view on the limited methodological novelty.

---

> > > > ### Author Response · Authors · 2024-11-27
> > > >
> > > > Dear Reviewer hRNb,
> > > >
> > > > Thank you for your thoughtful response and clarification. We again apologize for exceeding the page limit and appreciate your understanding.
> > > >
> > > > > ... I agree that exploring their application in the machine unlearning context is a valuable contribution. However, I see this value as being more applicative than methodological for the machine learning research community. With these considerations in mind, I have updated my score to reflect the significance of this application while maintaining my view on the limited methodological novelty.
> > > >
> > > > We’re pleased that you recognize our work as "a valuable contribution." In response to your comment about its contributions being more applicative than methodological, we’d like to highlight the following points that make our paper highly relevant to the ICLR community:
> > > >
> > > > 1. **Alignment with call for papers**: our work directly addresses the [ICLR's call for papers](https://iclr.cc/Conferences/2025/CallForPapers) on "societal considerations including fairness, safety, and privacy". By exposing critical security vulnerabilities in machine unlearning, we advance the understanding of risks in this emerging field.
> > > >
> > > > 2. **Methodological contributions**: our paper introduces the following methodological novelties:
> > > > - We propose novel defense strategies against the introduced attack (Section 4).
> > > > - We explore subset-selection-based attack strategies, extending beyond the input perturbation techniques commonly seen in poisoning literature (Section 4.2).
> > > > - We scale our attacks to challenging black-box and high-dimensional settings (Algorithm 3).
> > > >
> > > > 3. **Theoretical contributions**: our theoretical results (Appendix C) provide rigorous insights into the existence of the proposed attack, complementing our empirical findings and offering a strong foundation relevant to the machine learning community.
> > > >
> > > > We believe these contributions offer a blend of **methodological and theoretical insights**, complemented by the **practical implications** you have acknowledged, making our work highly relevant to the machine learning community.
> > > >
> > > > Thank you again for your constructive feedback and careful review.

---

### Official Review · Reviewer_jZsQ · 2024-11-04

**Soundness:** 2
**Presentation:** 3
**Contribution:** 2
**Rating:** 8
**Confidence:** 4

**Summary:**

This paper investigates the problem of adversarially degrading model accuracy by submitting adversarial unlearning requests for data not presented in the training set. Two settings are considered, white box and black box, depending on whether the attacker can access the model's gradients. Experiments show that such requests can significantly degrade the model's performance. The paper also presents the challenges of detecting the legitimacy of unlearning requests.

**Strengths:**

### Originality
* The idea of studying adversarial unlearning requests is novel and interesting. This is a valid concern when the model deployer accepts user-submitted unlearning data.
* The overall methodology is novel, including the use of the gradient-through-gradient technique, regardless of some inevitably similar techniques to poisoning attacks and gradient estimation in black-box attacks.

### Quality
* The methodology is solid under the proposed threat model, and correctly verified using experiments on CIFAR-10 and ImageNet, with three unlearning algorithms.
* The evaluations are generally comprehensive.

### Clarity
* The paper is well-written and easy to follow.

### Significance
* The overall threat model and takeaways are useful, under the assumption that it makes sense for a model deployer to accept user-submitted samples to execute the unlearning algorithm (see weakness for why this may not be the case).

**Weaknesses:**

### Originality

**Q1: Related work of security risks of unlearning tasks and poisoning attacks.**

Since this paper's threat model is similar in spirit to poisoning attacks, it's suggested to discuss more details on the related work and address two main questions. First, have there been prior efforts to study the security risks of machine unlearning tasks? Second, what's the difference between the threat models of this work and poisoning attacks, given that the attacker in this paper is also able to execute poisoning attacks from the beginning?

### Quality

**Q2: Practicalness of the threat model.**

The assumption of this paper is that the user may submit malicious images as unlearning requests to the model deployer, who will execute the unlearning algorithm on such images. However, I'm not sure if it ever makes sense for a model trainer to rely on user-submitted data to execute unlearning algorithms. In practice, it's usually the model provider's responsibility to identify which data belongs to the user and unlearn those data stored in the model provider's datastore. Can the authors provide some concrete use cases, preferably realistic, where any model provider will (1) not store the training data despite their original ability to train the model and (2) accept user-submitted samples to delete the data?

Extending from this concern, the proposed threat model may also cause some other issues, where User A may submit benign samples belonging to User B and request deletion on behalf of User B without User B's consent. I believe what's actually happening is that the model provider will go through their data store and identify those belonging to the requester, which they know for sure (1) belong to the requester and (2) were indeed used in the model training.

The second aspect of this question is that the attacker already controls a subset of the training data. In this case, why can't the attacker simply execute poisoning attacks, which are more stealthy than this attack? In this case, the attacker's identity is easy to trace, yet it's much harder to identify who has sent the poisoning samples.

**Q3: Regarding the defences with full access to training examples.**

This is similar to Q2 but applies specifically to Section 4.2. In this subsection, the defense has full access to the training samples, so I guess in this case the model provider does not have any limitations in Q2. If so, wouldn't the best defense be not trusting the user-provided data at all? As a model provider, or in fact, the service provider, they must have the ability to identify where the data was originally collected from. Otherwise, they won't be able to delete the user's data in their training set. Then they could just identify the precise set of user's data used for model training, and then execute the unlearning algorithm.

### Clarity

**Q4: Concerns regarding the experiment settings.**

1. In Section 3, the forget set has 10 to 100 samples. It's unclear how this range is devised, but 100 may sound too much IMO, as it means the model provider will accept up to 100 samples, which the requester claims belong to them, to update their model weights. It's suggested to provide more concrete justification for why such numbers are chosen.
2. The evaluation quantifies the attack's performance as the *maximum* accuracy degradation observed across a grid search of hyperparameters. I'm curious (and concerned) why the maximum instead of other more common metrics are reported, such as the average, medium, or confidence interval. It's also not very convincing that, in reality, the attacker would be able to have a grid search and obtain the best results.

### Significance

I think the overall idea and evaluation of executing machine unlearning algorithms on malicious inputs are interesting and useful. However, my main concern regarding the significance is that the threat model may not be realistic (Q2-Q3). It's true that unlearning algorithms have been implicitly assuming that the data to be deleted are indeed used for model training, but such an assumption is built upon another assumption that, in order to delete the user's data, the model provider must have the basic ability to identify the correct set of user data in their data store, and only then go to the next level of deleting data from the model. It's suggested to provide more justification for the practicalness of the proposed threat model, where the model provider is expected to delete data from the model without the ability to reliably identify the user's precise set of data stored in their database.

**Questions:**

See weaknesses.

---

> ### Author Response · Authors · 2024-11-21
> **Response to Reviewer jZsQ (Part 1)**
>
> Thank you for your feedback! We are glad to hear that you find our paper solid, well-written, and our evaluation generally comprehensive. We address your questions and comments below.
>
> **Q1: Related work of security risks of unlearning tasks and poisoning attacks**
> > have there been prior efforts to study the security risks of machine unlearning tasks?
>
> **A**: Thanks. As discussed in the related work section of our submission (Section 6), prior studies have explored various security risks in machine unlearning systems, including:
> - Unintended privacy leakage: Carlini et al. (2022), Hayes et al. (2024), Shi et al. (2024)
> - Vulnerability to re-introducing unlearned knowledge: Shumailov et al. (2024), Lucki et al. (2024)
> - Insufficient removal of poisoned data: Di et al. (2022)
>
> To the best of our knowledge, our work is the first to demonstrate that adversarial unlearning requests can cause catastrophic performance degradation in unlearned models.
>
> > what's the difference between the threat models of this work and poisoning attacks? attacker is also able to execute poisoning attacks
>
> **A**: You raised a valid point about poisoning attacks being a plausible alternative when the attacker has control over part of the training data. However, we’d like to make two important clarifications:
>
> - Our attack could operate under a slightly **weaker** threat model compared to poisoning attacks—it can be executed even without the attacker having control over a subset of the training data (Section 5.2).
> - Our attack identifies a new vulnerability **unique** to machine unlearning systems, whereas poisoning is a well-known attack vector applicable to all systems involving model training. These two types of attacks target different stages of a machine learning system, and thus our attack is **NOT** intended to compete with or replace standard poisoning attacks.
>
> Therefore, we view our contribution as orthogonal and possibly complementary to poisoning attacks.
>
> **Q2: Practicalness of threat model**
>
> > I'm not sure if it ever makes sense for a model trainer to rely on user-submitted data to execute unlearning algorithms. In practice, it's usually the model provider's responsibility to identify which data belongs to the user and unlearn those data stored in the model provider's datastore.
>
> **A**: We completely agree with the reviewer's opinion that the model provider has the responsibility to verify the unlearning requests, which is one of the main implications we are advocating in this paper. We note that while unlearning has been a promising research direction recently, it has not yet been widely deployed. Therefore, we do not have concrete examples of deployed systems to demonstrate this attack. However, all the mainstream formulation of unlearning has largely overlooked this problem. Our systematic study hopes to guide the evolution of unlearning protocols to address such vulnerabilities before widespread deployment.
>
> In the following, we address the two concrete questions raised by the reviewer:
>
> 1. *Why might a model provider not store all training data despite having originally trained the model?*
>
> There are two potential reasons:
> - **Storage limitations**: In cases of training on streaming data, the model deployer may only retain the most recent data due to limited storage capacity and have to discard older training data.
> - **Regulatory compliance**: regulations often require organizations to delete raw training data after a predefined retention period to minimize data exposure. Typically, the retention period for raw training data is shorter than that of the model.
>
> 2. *Why might a model provider accept user-submitted samples for unlearning instead of identifying them themselves?*
>
> Under the GDPR, individuals have the "right to be forgotten," which mandates that organizations remove personal data upon user request. In such cases, the model provider may rely on user-submitted samples to execute an unlearning operation.
>
> > the proposed threat model may also cause some other issues, where User A may submit benign samples belonging to User B and request deletion on behalf of User B without User B's consent.
>
> **A**: Thanks! While the scenario you described is orthogonal to our primary contribution and can be covered by the current attack framework, it is an interesting point. We will include a discussion of this in the final version.
>
> > I believe what's actually happening is that the model provider will go through their data store and identify those belonging to the requester, which they know for sure (1) belong to the requester and (2) were indeed used in the model training.
>
> **A**: Strong defenses like data ownership verification, as you suggested, could indeed prevent certain attacks. However, implementing them in practice is challenging  because privacy regulations typically require data to be anonymized, making it sometimes prohibitive to store user IDs or similar identifiers that link data to specific users.

---

> ### Author Response · Authors · 2024-11-21
> **Response to Reviewer jZsQ (Part 2)**
>
> **Q3: defences with full access to training examples**
> > If so, wouldn't the best defense be not trusting the user-provided data at all?... Then they could just identify the precise set of user's data used for model training, and then execute the unlearning algorithm.
>
> **A**:  We appreciate the comment. We kindly note that your suggestion of not trusting user-provided data is exactly the defense strategy we explored in Section 4.2 (the paragraph titled "Similarity searching and indexing"). Specifically, we examined a defense mechanism where the model deployer could compare submitted inputs against stored training examples to identify and unlearn them. However, we demonstrate that this approach still remains vulnerable to the indexing attacks: In such attacks, an adversary can strategically select a subset of valid training examples that, when removed, significantly degrades model performance (see Figure 5).
>
> **Q4: Experiment settings**
> > In Section 3, the forget set has 10 to 100 samples. It's unclear how this range is devised, but 100 may sound too much IMO, as it means the model provider will accept up to 100 samples, which the requester claims belong to them, to update their model weights. It's suggested to provide more concrete justification for why such numbers are chosen.
>
> **A**: The range of forget set sizes was selected to capture a spectrum of unlearning requests, from small (10 samples) to relatively large (100 samples).
>
> Regarding the concern about the size of 100 samples, we note that this is not unusually large in the context of prior work. The table below highlights the largest forget set sizes used in previous studies that use CIFAR-10 as a benchmark for unlearning:
>
> | Study                                                                           | Largest forget set size                     |
> |---------------------------------------------------------------------------------|---------------------------------------------|
> | Eternal sunshine of the spotless net: Selective forgetting in deep networks [1] | 5000                                        |
> | Towards adversarial evaluations for inexact machine unlearning [2]              | 4000                                        |
> | Machine Unlearning Competition, NeurIPS 2023 [3]                                | 5000 (10% of the CIFAR-10 training dataset) |
> | Towards Unbounded Machine Unlearning [4]                                        | 25 (small-scale), 5000 (large-scale)        |
>
> Compared to these studies, our choice of a maximum of 100 samples is relatively modest.
>
> > The evaluation quantifies the attack's performance as the maximum accuracy degradation observed across a grid search of hyperparameters. I'm curious (and concerned) why the maximum instead of other more common metrics are reported, such as the average, medium, or confidence interval.
>
> **A**: Our metric reports the **average** (across five randomly chosen forget sets) of the **maximum** accuracy degradation under attack. Using the "average" ensures statistical rigor, while the "maximum accuracy degradation" highlights the worst-case scenario, which is critical for evaluating system robustness. We argue that from a model deployer's perspective, focusing on non-optimal attack hyperparameters is less informative, as it fails to capture the most damaging potential outcomes.
>
> > It's also not very convincing that, in reality, the attacker would be able to have a grid search and obtain the best results.
>
> **A**: While online grid search may seem computationally intensive, we note that attackers can feasibly conduct **offline** hyperparameter optimization. In our implementation, we optimized hyperparameters for each forget set size during an initial search and reused them for all subsequent attacks of that size. This simulates a realistic attack scenario, where an adversary balances computational cost with maximizing effectiveness.
>
>
> **References**:
>
> [1] Golatkar, Aditya, Alessandro Achille, and Stefano Soatto. Eternal sunshine of the spotless net: Selective forgetting in deep networks. CVPR 2020
>
> [2] Goel, Shashwat, et al. Towards adversarial evaluations for inexact machine unlearning. arXiv preprint arXiv:2201.06640 (2022)
>
> [3] NeurIPS 2023 Machine Unlearning Challenge. https://unlearning-challenge.github.io/
>
> [4] Kurmanji, Meghdad, et al. Towards unbounded machine unlearning. NeurIPS 2024

---

> > ### Author Response · Authors · 2024-11-25
> > **Follow-up**
> >
> > Dear Reviewer,
> >
> > Thanks again for your detailed comments and suggestions. Since the rebuttal period is approaching its end, we would appreciate learning whether our responses have addressed your concerns. We are also happy to engage in further discussions.

---

> > > ### Comment · Reviewer_jZsQ · 2024-11-26
> > >
> > > Thanks for the detailed response. Below are my remaining concerns.
> > >
> > > **Q2: Thraet model.**
> > >
> > > I'm generally okay with the threat model's justification but would recommend being precise when introducing a new threat model. I agree with the statement that ML unlearning algorithms haven't discussed scenarios where D-forget does not belong to D-train. But this does not necessarily mean that such algorithms assume taking user-provided inputs to construct D-forget. If the latter is a given, then the proposed concerns would totally make sense. This means the paper's motivation is better decoupled into two steps:
> > > 1. There is a lack of discussion regarding whether ML unlearning algorithms should accept user-provided inputs to construct D-forget.
> > > 2. In scenarios where the unlearning algorithms have to take user inputs, two types of adversarial inputs can happen: (1) adversarially chosen benign D-forget and (2) adversarial examples present in D-forget.
> > >
> > > I think the author's response should be able to address the 1st point, which was the root cause of confusion. It's thus recommended to refine the motivation regarding this gap.
> > >
> > > **Q3: Section 4.2**
> > >
> > > I double-checked the results and the results are somewhat confusing. IIRC, it shows that the attacker can adversarially select 10 samples to construct the D-forget, which will reduce 30% accuracy if executed by the model deployer. I wonder where these 10 samples were selected and if there were additional assumptions. Were they selected from the entire training set (which the attacker should not have access to), or a chosen portion of the training set that the attacker initially provided for model training?
> > >
> > > **Q4: max vs. average**
> > >
> > > Since this paper is an attack paper, the results are higher the better. So it would make more sense to report the mean and std of the attack's bare effectiveness across several runs. Only in this way, we can understand the true distribution of the attack's performance. The hyper-parameter tuning should belong to a separate ablation study to show how the attack's performance is sensitive to these factors, and the possibility & stability of offline hyper-parameter optimization.

---

> ### Author Response · Authors · 2024-11-28
>
> Dear Reviewer jZsQ,
>
> Thank you for your thoughtful response and follow-up questions.  We address them as follows.
>
> **Q2: Threat model**
>
>
> > I'm generally okay with the threat model's justification but would recommend being precise when introducing a new threat model...I think the author's response should be able to address the 1st point, which was the root cause of confusion. It's thus recommended to refine the motivation regarding this gap.
>
> We’re glad to hear that our clarification about the threat model largely addressed your concern. We also agree with your suggestion to further clarify our motivation, particularly by discussing the two mentioned points.
>
> We plan to restructure the introduction flow of the paper as follows: While machine unlearning algorithms have been proposed, there has been limited discussion on their deployment, particularly regarding whether these algorithms should accept user-provided inputs to construct D-forget. In this paper, we highlight two potential vulnerabilities that attackers could exploit if user-provided inputs are allowed to construct D-forget:
> - Adversarially perturbed examples from D-forget (Section 3), which can be mitigated by certain defenses (Section 4.1).
> - Adversarially chosen valid examples from D-forget, which pose a greater challenge for the deployer to defend against (Section 4.2).
>
> We are happy to update the paper accordingly if this revised flow makes sense to you.
>
>
>
>
>
> **Q3: Section 4.2**
>
> > I double-checked the results and the results are somewhat confusing. IIRC, it shows that the attacker can adversarially select 10 samples to construct the D-forget, which will reduce 30% accuracy if executed by the model deployer. I wonder where these 10 samples were selected and if there were additional assumptions. Were they selected from the entire training set (which the attacker should not have access to), or a chosen portion of the training set that the attacker initially provided for model training?
>
> Thank you for your question. In this setting, the attacker adversarially selects N examples to construct D-forget. Figure 5 illustrates that for N=10, this results in the model achieving ~30% classification error after unlearning (compared to only 3.8% classification error for an average-case forget set).
>
> We note that this is a proof-of-concept experiment, where we allow the attacker to exploit its maximal power by randomly sampling N images *from the entire training set*. For your reference, we also conducted experiments where the attacker has access to **only a portion of the data**. Below are the results for N=10:
>
>
> | Attacker's portion to training data | Max Retain Error (%) |
> |---------|----------------------|
> | 5%      | 21.6                |
> | 10%     | 26.8                |
> | 20%     | 27.3                |
> | 50%     | 28.7                |
> | 100%    | 30.9                 |
>
> We are happy to include these additional results for different values of  N's in the final version of the paper.
>
>
> **Q4: max vs. average**
> > Since this paper is an attack paper, the results are higher the better. So it would make more sense to report the mean and std of the attack's bare effectiveness across several runs.
>
> Thank you for your follow-up question. We’ve included the average-case results for the main tables (Table 1 and Table 4) in Appendix B.3 for your reference.

---

> > ### Comment · Reviewer_jZsQ · 2024-11-29
> >
> > Thanks for the response. I think the revised flow makes much more sense now.
> >
> > The updated results for Q3 make sense now. However, it's worth pointing out that the attacker in Sec 4.2 is much stronger than the main threat model, as they control more than the samples they submitted. A more precise modeling of the attacker would be that they can only sample the N points from their controlled part of the data -- where 5% would still be unusually high, as in Tabs 3 and 4 the attacker only had control of D-forget with 10 - 100 samples.
> >
> > I think it's acceptable to increase the attacker's capability to quantify the defender's hardness, but it may have overestimated the damage caused by the attacker if the paper wants to make a point on the attack's side (I understand it's in a defense section). I would recommend clarifying the gap in the threat model and focusing on the defense's side here, and being conservative when discussing the attacker's capability.
> >
> > Overall I think the paper should be in good shape with all the above updates implemented. I've adjusted my score accordingly.

---

> > > ### Author Response · Authors · 2024-11-30
> > >
> > > Dear Reviewer jZsQ,
> > >
> > > We sincerely appreciate your constructive feedback and for raising your score.
> > >
> > > We’re happy to hear that you find the revised flow clearer. We’ll include this update in the final paper (unfortunately, it seems we can’t update the PDF submission at this stage).
> > >
> > > We also agree with your point that the results in Section 4.2 may appear stronger than those under the main threat model. In the final version, we’ll revise the main results to focus on scenarios where the attacker has access to very few examples (e.g., $\leq$ 5%), while keeping the results for cases with greater access for reference.
> > >
> > > Thank you again for your feedback.

---

### Official Review · Reviewer_Hn2d · 2024-11-04

**Soundness:** 3
**Presentation:** 4
**Contribution:** 3
**Rating:** 8
**Confidence:** 3

**Summary:**

This paper explores security risks within machine unlearning schemes, a topic gaining importance as the demand for ensuring "the right to be forgotten" grows. Specifically, the paper introduces a type of attack that degrades a model's accuracy by submitting adversarial unlearning requests, with evaluations demonstrating its effectiveness. Additionally, the paper shows that various existing verification mechanisms fail to detect these proposed attacks.

**Strengths:**

- The concept of adversarial machine unlearning attacks is innovative and contributes a new perspective to the field.
- In addition to empirical evaluations, the paper provides a theoretical demonstration of the proposed attack, enhancing its depth and rigor.
- The paper is well-structured, with clear and accessible content that is easy to follow.

**Weaknesses:**

Overall, I enjoyed reading this paper. Below are some suggestions that could help improve it further:

- *Scalability Limitations in Black-Box Attacks*: While the paper’s black-box attacks yield promising results, they might face challenges when applied to high-dimensional data. Including a more detailed discussion of this limitation could add depth and insight. Addressing these challenges would provide valuable context about the trade-offs and bottlenecks in black-box attacks, enhancing the practical understanding for researchers and practitioners in the field of machine unlearning and security.
- *Limited Exploration of Adaptive Defense Mechanisms*: This paper makes a valuable contribution by assessing the effectiveness of various defensive mechanisms and highlighting potential vulnerabilities. However, investigating additional adaptive defenses, such as anomaly detection or similarity-based clustering methods, could strengthen this section.
- *Ambiguity in the definition of “Black-Box” Attack*: Although termed a “black-box” attack, the method used by this paper still requires access to the training loss. This might be somewhat misleading, as traditional black-box adversarial attacks are generally assumed to operate without any internal information about the target model. Adjusting this terminology or clarifying the scope could improve precision and align expectations.

**Questions:**

- Can you think about possible ways to enhance the defense against such adversarial machine unlearning attacks?

---

> ### Author Response · Authors · 2024-11-21
> **Response to Reviewer Hn2d**
>
> We appreciate your feedback and are glad to hear you enjoyed our paper. We address your questions and comments below.
>
> **Q1: Scalability Limitations in Black-Box Attacks**
>
> > While the paper’s black-box attacks yield promising results, they might face challenges when applied to high-dimensional data. Including a more detailed discussion of this limitation could add depth and insight. Addressing these challenges would provide valuable context about the trade-offs and bottlenecks in black-box attacks, enhancing the practical understanding for researchers and practitioners in the field of machine unlearning and security.
>
> **A**: We appreciate the comment. From a theoretical perspective, the gradient estimators in the black-box setting can have larger variance in the high-dimensional setting, and our experiments confirm that attacking high-dimensional data (e.g., ImageNet) is more challenging (Table 2). Increasing the number of estimators and averaging them could partially mitigate this issue. As suggested, we have included this discussion in the revised PDF.
>
> **Q2: Limited Exploration of Adaptive Defense Mechanisms**
>
> > This paper makes a valuable contribution by assessing the effectiveness of various defensive mechanisms and highlighting potential vulnerabilities. However, investigating additional adaptive defenses, such as anomaly detection or similarity-based clustering methods, could strengthen this section.
>
> **A**: Thank you for the comment. We note that Section 4 already explores several anomaly detection approaches, including hashing-based, embedding-based, and pixel distance-based methods. If you have specific suggestions for additional defenses, we would be happy to consider and discuss their incorporation.
>
> **Q3: Ambiguity in the definition of “Black-Box” Attack**
> > Although termed a “black-box” attack, the method used by this paper still requires access to the training loss. This might be somewhat misleading, as traditional black-box adversarial attacks are generally assumed to operate without any internal information about the target model. Adjusting this terminology or clarifying the scope could improve precision and align expectations.
>
> **A**: Thank you for this valuable feedback. In our setup, we define the black-box attack as one that requires access to the training loss. We acknowledge that this differs from the stricter definition, where the attacker has access only to the model's outputs. However, similar setups have also been referred to as "black-box" in prior literature [1, 2, 3]. We have clarified this in the revised PDF as suggested (lines 135-136).
>
> **Q4: Potential defenses**
> > Can you think about possible ways to enhance the defense against such adversarial machine unlearning attacks?
>
> **A**: Thank you for the comment. As noted in Section 4.2, the most robust defense we can propose is for the model deployer to avoid directly using user-submitted inputs for unlearning (the paragraph titled with “Similarity searching and indexing”). Instead, they could compare the submitted inputs to already stored examples and use these for unlearning. However, this defense requires the model deployer to pre-store such examples and still remains susceptible to indexing attacks, where an attacker selects a subset of valid training examples that degrade model performance after unlearning.
>
> **References**:
>
> [1] Chen, Pin-Yu, et al. Zoo: Zeroth order optimization based black-box attacks to deep neural networks without training substitute models.
>
> [2] Tu, Chun-Chen, et al. Autozoom: Autoencoder-based zeroth order optimization method for attacking black-box neural networks.
>
> [3] Liu, Yiyong, et al. Membership inference attacks by exploiting loss trajectory.

---

> > ### Comment · Reviewer_Hn2d · 2024-11-21
> >
> > Thank you for the clarification and the revision.

---

### Meta-Review · Area_Chair_zTh2 · 2024-12-23

**Metareview:**

This paper received ratings of 8,8,3. The paper presents promising results on black-box attacks and machine unlearning but has several key weaknesses. It overlooks scalability challenges for high-dimensional data in black-box attacks, and fails to fully explore adaptive defense mechanisms like anomaly detection. The definition of “black-box” attack is ambiguous, as the method still requires access to training loss, which deviates from traditional black-box assumptions. Additionally, the threat model may not be realistic, as it assumes model providers accept user-submitted data for unlearning, which is uncommon in practice. There are also concerns about the model provider’s ability to reliably identify user data for deletion and the practicality of the attack in real-world scenarios. The evaluation method, including the use of maximum accuracy degradation, is questioned for its relevance and realism. Lastly, the paper would benefit from a more detailed exploration of related work on the security risks of machine unlearning and poisoning attacks.

**Additional Comments On Reviewer Discussion:**

The rebuttal addressed most issues raised by the reviewers. One reviewer is still rejecting it. The main weakness reported by the reject reviewer is: I see the value of this paper as being more applicative than methodological for the machine learning research community. That said, it may hold greater relevance and value for the security community, where its implications could be further explored and appreciated.

The AC believes the submission has sufficient positive aspects so can be accepted.

---

### Decision · Program_Chairs · 2025-01-22

Accept (Poster)